# Lineage-defined leiomyosarcoma subtypes emerge years before diagnosis and determine patient survival

Nathaniel D. Anderson [1,2], Yael Babichev[3,17], Fabio Fuligni[4,17], Federico Comitani [1], Mehdi Layeghifard [1], Rosemarie E. Venier [3], Stefan C. Dentro[5], Anant Maheshwari[1], Sheena Guram[3], Claire Wunker[3,6], J. Drew Thompson [1], Kyoko E. Yuki[1], Huayun Hou [1], Matthew Zatzman[1,2], Nicholas Light[1,6], Marcus Q. Bernardini[7,8], Jay S. Wunder [3,9,10], Irene L. Andrulis [2,3,11], Peter Ferguson[7,9,10], Albiruni R. Abdul Razak[7], Carol J. Swallow[3,6,9,12], James J. Dowling[1,11], Rima S. Al-Awar[13,14], Richard Marcellus[13], Marjan Rouzbahman[2,7], Moritz Gerstung [5], Daniel Durocher [3,11], Ludmil B. Alexandrov [15], Brendan C. Dickson [2,3,16], Rebecca A. Gladdy [3,6,9,12 ✉] & Adam Shlien [1,2,4 ✉]

Leiomyosarcomas (LMS) are genetically heterogeneous tumors differentiating along smooth muscle lines. Currently, LMS treatment is not informed by molecular subtyping and is associated with highly variable survival. While disease site continues to dictate clinical management, the contribution of genetic factors to LMS subtype, origins, and timing are unknown. Here we analyze 70 genomes and 130 transcriptomes of LMS, including multiple tumor regions and paired metastases. Molecular profiling highlight the very early origins of LMS. We uncover three specific subtypes of LMS that likely develop from distinct lineages of smooth muscle cells. Of these, dedifferentiated LMS with high immune infiltration and tumors primarily of gynecological origin harbor genomic dystrophin deletions and/or loss of dystrophin expression, acquire the highest burden of genomic mutation, and are associated with worse survival. Homologous recombination defects lead to genome-wide mutational signatures, and a corresponding sensitivity to PARP trappers and other DNA damage response inhibitors, suggesting a promising therapeutic strategy for LMS. Finally, by phylogenetic reconstruction, we present evidence that clones seeding lethal metastases arise decades prior to LMS diagnosis.

[1] Program in Genetics and Genome Biology, The Hospital for Sick Children, Toronto, ON, Canada. [2] Department of Laboratory Medicine and Pathobiology, University of Toronto, Toronto, ON, Canada. [3] Lunenfeld-Tanenbaum Research Institute, Mount Sinai Hospital, Toronto, ON, Canada. [4] Department of Pediatric Laboratory Medicine, The Hospital for Sick Children, ON Ontario, Canada. [5] European Molecular Biology Laboratory, European Bioinformatics Institute, Hinxton, UK. [6] Institute of Medical Science, University of Toronto, Toronto, ON, Canada. [7] Princess Margaret Cancer Centre, University Health Network, Toronto, ON, Canada. [8] Department of Obstetrics and Gynaecology, University of Toronto, Toronto, ON, Canada. [9] Department of Surgery, University of Toronto, Toronto, ON, Canada. [10] University Musculoskeletal Oncology Unit, Mount Sinai Hospital, Toronto, ON, Canada. [11] Department of Molecular Genetics, University of Toronto, Toronto, ON, Canada. [12] Division of General Surgery, Mount Sinai Hospital, Toronto, ON, Canada. [13] Drug Discovery Program, Ontario Institute for Cancer Research, Toronto, ON, Canada. [14] Department of Pharmacology and Toxicology, University of Toronto, Toronto, ON, Canada. [15] Department of Cellular and Molecular Medicine and Department of Bioengineering and Moores Cancer Center, University of California, San Diego, La Jolla, CA, USA. [16] Department of Pathology and Laboratory Medicine, Mount Sinai Hospital, Toronto, ON, Canada. [17] These authors contributed equally: Yael Babichev, Fabio Fuligni. ✉email: gladdy@lunenfeld.ca; adam.shlien@sickkids.ca

Leiomyosarcoma (LMS) is a malignant neoplasm of smooth muscle differentiation, which accounts for 10–20% of sarcoma diagnoses[1]. The heterogeneity of its site of origin and clinical course, including the development of metastasis and response to therapy, makes the treatment of LMS particularly challenging. Common sites of presentation include the extremities, the uterus, and the abdomen, the latter of which predominately arises from the retroperitoneum, but also from the viscera. While all LMS, regardless of anatomical origin, differentiate along the smooth muscle lineage, the disease site continues to be used to stratify care pathways. Rates of metastasis at 10 years vary by disease site (31% in extremity, 58% in the abdomen, 53–71% in the uterus)[2,3]. However, it is unknown whether the molecular factors leading to these site-specific survival differences are inherent to the lineage, differentiation status or mutational processes operative in the cells of origin from which the disease develops, or are acquired, involving unique driver gene mutations.

As the primary pattern of failure is the development of metastasis, outcomes for LMS patients have not improved as the overall effectiveness of current systemic treatment remains poor[4]. Recent studies using targeted sequencing and/or RNA-seq have confirmed frequent alterations of *TP53*, *RB1*, *PTEN*, and *ATRX*, as well as recurrent amplifications of 17p11.2-p12 (*MYOCD*)[5,6]. One of these studies also suggested that LMS may be sensitive to poly (ADP-ribose) polymerase inhibition (PARPi) due to acquired mutations in *BRCA1/BRCA2*. The "BRCAness" mutational signature was reported in 48/49 (98%) patients in this study[5]. If validated, this would support the use of PARPi in LMS. Whole-genome sequencing (WGS) typically yields ~100X more mutations than exome sequencing and therefore, is the preferred approach for validating and refining mutational signature analyses, especially for separating related signatures from each other[7].

The majority of LMS patients present with non-specific symptoms at 50–60 years of age. These tumors may develop gradually over the course of several years. A better understanding of the timing of mutations in LMS would therefore provide knowledge about tumor development, and perhaps aid in early detection. Similarly, understanding the ongoing mutational dynamics of LMS, including its mutation rate and overall model of tumor evolution, could define rational surveillance strategies for patients prior to the development of metastatic spread.

In this work, we study the mutational processes underlying primary and metastatic LMS by analyzing 70 whole genomes and 130 transcriptomes from 113 patients. This includes samples from multiple areas from a single tumor and/or distant metastatic relapses separated in time. We observe widespread genetic diversity within primary tumors and between metastatic relapses, especially with respect to rearrangements and clustered mutations that are only detectable by WGS. The results provide compelling support for molecular subtyping in LMS, and evidence for early systemic spread years prior to diagnosis. This study also highlights the potential use of a DNA damage inhibitors (DDRi), including PARP trappers, as a promising therapeutic avenue in these patients.

## Results

### Whole-genome and transcriptome sequencing of a unique LMS cohort.
To reconstruct the molecular events underpinning LMS, we carried out an analysis of the patterns, location, and evolution of somatic mutations in LMS, both between and within tumors. In total, 70 genomes and 130 transcriptomes of LMS were analyzed (Supplementary Fig. 1A). Of these, 53 samples from 34 patients were newly genome sequenced. An additional 18 whole genomes, previously reported by The Cancer Genome Atlas

(TCGA)[6], were included. Only validated LMS samples approved by the TCGA-SARC program were used (Supplementary Fig. 1B, see Methods). Matched-blood or tissue was used as a reference for all patients. We conducted a detailed pathological review of all specimens including those from TCGA. During this review, we detected a pathogenic *KIT* variant known to be associated with gastrointestinal stromal tumors[8] (GIST) in one TCGA LMS sample, prompting its removal from further analysis. All genomes were processed through the same established informatics pipeline[9] to detect somatic mutations, including substitutions, small insertions or deletions (indels), copy number changes, structural rearrangements as well as clustered and complex events (Supplementary Data 1–6). For seven patients, we sequenced multiple areas from a single tumor and/or distant metastatic relapses separated in space or time. Sixteen informative samples from five of these patients were selected for additional, targeted, deep sequencing (~700X) to validate substitutions, indels, and structural variants present in these multiregion or matched primary-metastatic samples (Supplementary Fig. 2). Full RNA sequencing analysis for expression and fusion gene analysis was also carried out for 130 (51 Toronto, 79 TCGA) transcriptomes. To our knowledge, this work represents the broadest analysis of somatic mutations in LMS to date.

### Three molecular subtypes of LMS, with distinct genomes and transcriptional programs, correlate with patient survival.
Although labeled as a single disease characterized by smooth muscle differentiation, the clinical presentation and behavior of LMS are highly variable[10]. The degree to which this variability is explained by overall genomic and transcriptomic features is largely unknown. As a starting point, we examined the tumors' whole transcriptomes and found three predominant gene expression subtypes by principal component analysis that were not fully accounted for by disease site (Fig. 1a). We observed three LMS subtypes consistent with previous studies[5,6,11–14] that harbored established gene expression markers of survival (Supplementary Fig. 3A). Subtype 1 was mixed: 43.4% gynecological, 30.4% extremity, 13.0% abdominal, 8.6% metastases, and 4.3% other. Subtype 2 largely consisted of abdominal and to a lesser degree extremity tumors (63.5% abdominal, 17.6% extremity, 1.1% gynecological, 15.2% metastases, and 2.3% other), while subtype 3 was composed almost exclusively of gynecological LMS (90.9%, 9.1% other). The transcriptome-defined subtypes were also associated with differences in genome-wide mutation burdens (Fig. 1b, Supplementary Data 7). LMS subtypes 1 and 3 (mixed and gynecological, respectively) harbored a higher overall burden of somatic mutations and displayed inferior overall and disease-specific survival compared to subtype 2 (abdominal/extremity; Supplementary Fig. 3B, C). This was true for all classes of mutations, including substitutions (2.7 vs 1.9 Mut/Mb, n.s.), indels (605 vs 359 v indels, $p < 0.05$, Welch's t-test), and large-scale rearrangements (224 vs 82 rearrangements, $p < 0.01$, Welch's t-test).

We then validated these expression subtypes using an orthogonal approach based on Uniform Manifold Approximation and Projection (UMAP[15]) and Density-Based Spatial Clustering of Applications with Noise (DBSCAN[16]; see Methods). To optimize subtype generation and classification, we took two experimental approaches. First, we included 12,419 additional tumors of various histologies in our analysis. Second, we clustered the 130 LMS tumors with 1735 normal tissues from cancer-free donors[17,18]. We found the same subtypes using our pan-cancer cluster approach, except we were now able to divide subtype 2 into two additional classes: subtype 2a, which is comprised almost exclusively of abdominal LMS, and subtype 2b, which consists of abdominal and extremity tumors.

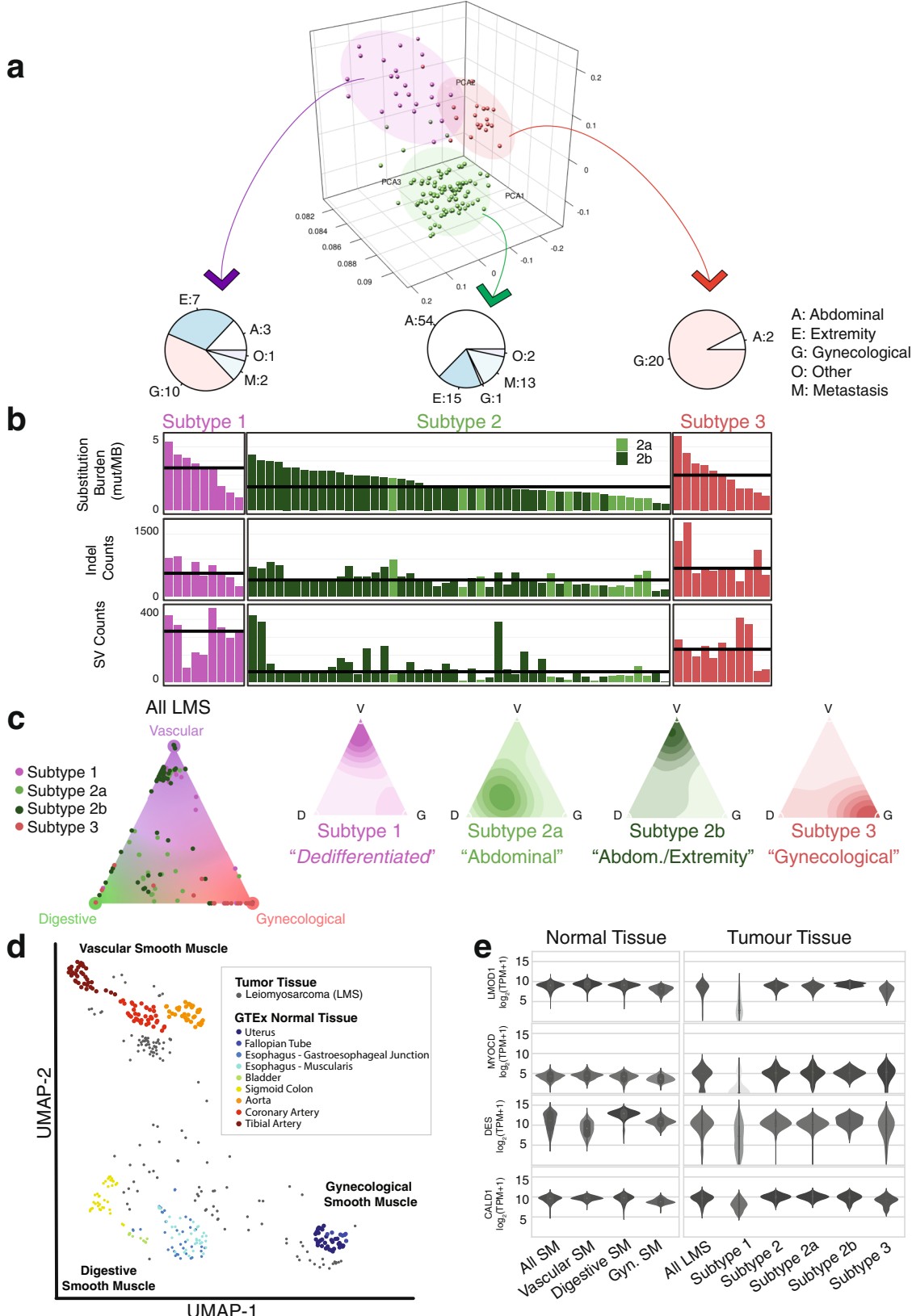

(Supplementary Fig. 4). Out of the 130 RNA samples, three cases (2.3%) clustered apart from the others––falling outside the three LMS subtypes. We reviewed the histopathology from each case but did not find any further evidence to support a change in diagnosis. In one instance, a gastric LMS resembled stomach

cancers from TCGA. Differences in the tissue of origin may explain these few exceptional LMS tumors. In our second approach, matching tumors to normal tissues, we found that three main tissue types strongly resembled LMS (out of 51 types): (1) Digestive smooth muscle (esophagus muscularis/

**Fig. 1 Genomic differences and normal cellular lineages of LMS transcriptional subtypes. a** Principal Component Analysis (PCA) of LMS transcriptomes ($n = 79$ TCGA RNA-seq and $n = 51$ Toronto RNA-seq) leads to three defined subtypes of LMS, in which segregation is broadly influenced by anatomical location. While subtype 1 is a mix of all sites (extremity, abdominal and gynecological), subtype 2 is largely abdominal, with some extremity tumors. Subtype 2 can be further sub-stratified into subtype 2a and 2b. The metastatic lesions in subtype 2 cluster with their matched primary tumors. There are two metastatic tumors in subtype 1 that are from the same patient. Lastly, subtype 3 represents a largely gynecological (uterine, vaginal, fallopian tube) subtype. **b** Genomic point mutation, indel (insertion/deletion), and structural variant (SV) burdens are lower in subtype 2 than subtypes 1 or 3. Horizontal black lines represent the median values for each subtype. Source data are provided in Supplementary Data 7. **c** LMS molecular subtypes are of distinct smooth muscle lineages: vertices of the triangular plot represent smooth muscle of vasculature, digestive tissue, and gynecological tissue. Individual dots represent LMS cancers and where they lie in the cluster. Adjacent contour plots illustrate density distribution of LMS molecular subtypes. **d** Uniform Manifold Approximation and Projection (UMAP) illustrates clustering of 271 muscle-related GTEx normal tissue types (from the Genotype-Tissue Expression Program) and 130 LMS reveals distinct smooth muscle lineages of LMS subtypes. **e** Boxplots represent the expression (in transcripts per million, TPM), for smooth muscle (SM) genes: LMOD1 (leiomodin 1), MYOCD (myocardin), DES (desmin), and CALD1 (caldesmon) are key smooth muscle genes that are commonly expressed in LMS. The boxes represent the 25th and 75th percentile (bottom and top of box), and median value (horizontal band). The whiskers indicate the variability outside the upper and lower quartiles. These genes are highly expressed in vascular ($n = 110$), digestive ($n = 119$), and gynecological (Gyn., $n = 42$) normal smooth muscle. Genes are also expressed in subtype 2 ($n = 85$) and subtype 3 ($n = 22$) LMS, but not as highly in LMS subtype 1 ($n = 23$).

---

gastroesophageal junction, colon, bladder); (2) vascular smooth muscle (tibial/coronary arteries, aorta); and (3) gynecological/fallopian tube smooth muscle (Fig. 1c, d, Supplementary Fig. 5). As expected, gynecological LMS most resembled normal gynaecological tissue. Subtype 2a (almost exclusively abdominal) clustered with digestive smooth muscle, while subtype 1 (mixed) and 2b (abdominal/extremity) mostly resembled vascular smooth muscle, consistent with extremity and retroperitoneal LMS being associated with vasculature. These analyses define the cellular lineage underlying the LMS molecular subtypes.

**Genomic alterations in muscle-related genes segregate in LMS subtypes.** Having seen how closely the molecular subtypes of LMS adhere to their respective smooth-muscle lineages and knowing certain muscle marker genes are overexpressed in some LMS[11,12], we next searched for alterations in genes related to smooth-muscle differentiation and function. We looked for mutations (point mutations, indels, structural variants, or focal copy number changes) in smooth muscle marker genes: LMOD1, CALD1, MYOCD, DMD, ACTG2, DES, CFL2, and SLMAP. The most frequent alterations detected were recurrent intragenic deletions of the dystrophin (DMD) gene in 8/51 (16%) and focal myocardin (MYOCD) amplifications in 20/51 (39%) patients' genomes (Supplementary Fig. 6A–C). In fact, DMD deletions were as frequent as ATRX deletions and among the most recurrent structural variant changes in LMS, after RB1 (Supplementary Data 3). Intriguingly, DMD deletions occur predominantly in LMS subtype 1, and to a lesser extent subtype 3, while MYOCD amplifications occur more commonly in subtypes 2 and 3 (Supplementary Fig. 6d). Of note, MYOCD (17p12) amplifications frequently accompanied TP53 (17p13) losses in a joint segmental gain/loss pattern (Supplementary Fig. 6B), and in five cases, MYOCD was amplified via complex copy number patterns (Supplementary Fig. 6C). With respect to DMD alterations, it is well-established that dystrophin deficiency causes Duchenne and Becker muscular dystrophies; however, its role in cancer is not well defined. Recent data have emerged suggesting that dystrophin can act as a tumor suppressor and limits metastasis in myogenic cancers[19]. Furthermore, it was reported that DMD deletions, that were detected by SNP array, occurred in 3/7 primary LMS and 8/13 metastatic LMS that abrogated expression of full-length dystrophin transcript, which encodes Dp427m[19]. To investigate this further, we searched for the loss of expression of the Dp427m transcript in samples where a genomic DMD deletion was detected and found either no expression or lower expression of its transcript in 7/8 (88%) cases (Supplementary Fig. 7A). Surprisingly, and irrespective of molecular subtype, the

majority of gynecological LMS (21/28, 75%) had a reduced or complete lack of expression of the Dp427m transcript even when a DMD deletion was not detected, suggesting that there may be alternative mechanisms of dystrophin inactivation. This appears to be a unique feature of gynecological LMS that is not seen in soft-tissue LMS without a DMD deletion or normal uterine tissues (Supplementary Fig. 7B). To confirm that the dystrophin isoforms seen in LMS resemble those seen in Duchenne muscular dystrophy (DMD) patients, we RNA sequenced muscle biopsies from six patients with DMD and observed that DMD in dystrophin-deleted LMS transcriptomes do have similar expression to muscular dystrophy patients (Supplementary Fig. 7A). Taken together, we find frequent alterations in the DMD gene and loss of full-length DMD expression associated with LMS subtypes 1 and 3 that are consistent with inferior outcomes.

**Subtype 1 LMS are associated with myogenic dedifferentiation and high immune infiltration.** Given the prevalence of deletions of dystrophin––a critical muscle protein––in subtype 1, we wondered if subtype 1 constituted an overall less differentiated form of LMS, which is known to be a prognostic marker of poor outcome[20]. Consistent with this notion, most markers of muscle differentiation, including leimodin1 (LMOD1), caldesmon (CALD1), and MYOCD were diminished in subtype 1 (Fig. 1e, Supplementary Fig. 8). We then compared LMS to undifferentiated pleomorphic sarcoma (UPS), obtained from the TCGA, as these two entities can be difficult to distinguish histologically[21]. Most of the UPS tumors clustered with LMS subtype 1––the only subtype to have non-LMS cancers grouped with it (Supplementary Fig. 9). This suggests that LMS subtype 1 may represent an aggressive dedifferentiated form of LMS.

For many solid tumors, dedifferentiation towards 'stemness' is immunosuppressive[22,23], however, it is unclear how this relates to LMS and its molecular subtypes. To determine the relationship between the immune microenvironment and the state of dedifferentiation in LMS, we obtained leukocyte proportion and cell type for the 79 TCGA LMS samples[24] and performed an in silico immune analysis. Consistent with previous work[25–27], M2 macrophages were the most prevalent immune cell in LMS. Interestingly, we found that the dedifferentiated subtype 1 LMS harbored a higher leukocyte content, specifically M2 macrophages, than subtypes 2 or 3 (Supplementary Fig. 10A, B, see Methods). Of note, the established marker for this subtype, ARL4C expression, was overexpressed in subtype 1 LMS, as expected[11] (Supplementary Fig. 10C). Tumors with higher leukocyte fractions are known to be the most responsive to immune checkpoint inhibition, as recently demonstrated in UPS

and dedifferentiated liposarcoma[28]. In LMS, the infiltrate composition reflects an M2 macrophage-dominated, low lymphocytic profile that is consistent with an immunosuppressed tumor microenvironment. This may render LMS, and in particular poorly differentiated subtype 1 LMS, more responsive to macrophage-focused immunomodulatory agents[29,30]. However, these findings warrant further investigation.

**Genomic substitution and indel signatures identify prior therapy exposure and a treatment strategy for LMS.** Next, we performed a mutational signature analysis to determine if LMS subtypes are driven by different mutagenic processes. We began with an updated de novo extraction approach using an established signature extraction tool, SigProfiler[7,31] which differs from previous 'refitting' approaches used on LMS exomes[5]. Using SigProfiler for LMS mutational signatures, we optimized our approach for the increased number and variety of mutations detectable by WGS, including signatures based on single-base substitutions (SBS), double-base substitutions (DBS), as well as insertions and deletions (ID). Owing to the larger number of data points provided by WGS, previously correlated signatures could now be separated. One such signature is the HR-deficiency or 'BRCAness' signature (SBS3). We validated our power to discriminate SBS3 from other 'flat' signatures by using two approaches. First, we demonstrated a strong correlation (>99% Pearson coefficient) between our output and the ground-truth catalogue from ICGC/TCGA Pan-Cancer Analysis of Whole Genomes Network (Supplementary Fig. 11, see Methods). Second, we removed SBS3 from the reconstruction of the mutational profile and evaluated the changes in cosine similarity (Supplementary Fig. 12, see Methods). Thus, using a de novo extraction approach with WGS data points we can confidently identify specific intrinsic sources of mutation.

27 established signatures found in the COSMIC database were identified in this group of 70 LMS tumor genomes, including fourteen SBS, five DBS, and eight ID signatures (Fig. 2a, Supplementary Fig. 13). Of these, 13 have a known or proposed etiology. Four signatures of sequencing artifacts (SBS45, SBS46, SBS49, SBS52) were removed (see Methods). Two ID signatures and one DBS not found in COSMIC were also identified and are of unknown cause (Supplementary Fig. 14). Of note, ID8 was detected in every sample known to have been treated with radiation therapy (9/9), consistent with the suggested etiology of this signature. This included two cases where patients received prior radiation for previous cancer, suggesting LMS is secondary cancer in these individuals. Overall, mutational signatures highlighted the importance of DNA repair defects in LMS. We found SBS8, linked to a deficiency in nucleotide excision repair[32], as well as SBS3 and ID6, which are signatures of defective homologous recombination-based DNA damage repair[31,33]. Either SBS3 or ID6 was detected in 39/61 (63.9%) of LMS samples (13/65 SBS3 alone, 18/65 ID6 alone, and 8/61 in both). Thus, we are able to confirm the existence of HR-deficiency substitution signatures, albeit at a lower frequency than previously reported[5], as well as the existence of a corresponding HR-indel signature in LMS.

**Functional validation of DNA damage repair and homologous recombination deficiency (HRD) in LMS.** To validate if SBS3, ID6, and increased levels of DNA repair and damage pathway dysregulation identified in LMS primary tumors and metastasis represent a targeted therapeutic opportunity, we functionally tested a panel of LMS cell lines (five LMS primary lines, three ATCC) to evaluate their sensitivity to DNA damage and PARP inhibitors (Fig. 2b, c, Supplementary Fig. 15). These cell lines are reflective of

the diversity of LMS sites of origin, as lines were derived from abdominal, extremity, and gynecological sites (Supplementary Data 8). First, we tested the efficacy of other DDRi to inhibit LMS cell growth. These inhibitors target key kinases that play a pivotal role in the DNA damage response pathway. All LMS cell lines were responsive to the CHK1 inhibitor, Prexasertib HCl (LY2606368). Similarly, 7/8 LMS cell lines were responsive to the WEE1 inhibitor, Adavosertib (AZD1775), while only 5/8 LMS cell lines were partially or fully responsive to the ATR inhibitor, Ceralasertib (AZD6738) (Fig. 2b). To determine the effectiveness of PARPi in LMS, we compared LMS cell line responsiveness to primary UPS cell lines ($n = 5$, see Supplementary Data 9) and controls: (1) a CRISPR TP53 deletion (RPEΔp53, HR-intact) cell, (2) a TP53 and BRCA1 deletion (RPEΔp53ΔBRCA1, HR-deficient[34]), and (3) Hs 789.Sk. LMS cell lines were then treated with 24-point concentrations of talazoparib and olaparib (0.013–10 μM) in three independent experiments. All LMS cell lines were highly responsive to the PARP trapper, talazoparib, with a median $EC_{50}$ of 0.37 μM (range 0.04–0.8 μM), with the most potent inhibition occurring in gynecological LMS cell compared to soft tissue LMS cell lines (0.055 μM vs 0.51 μM) (Fig. 2b, c). In contrast, most LMS cell lines were not as responsive to olaparib, given our strict threshold criteria ($EC_{50} < 1$ μM is considered responsive). Taken together, it is evident that there are many promising therapeutic avenues for LMS patients that largely rely on LMS cells' inability to repair DNA efficiently.

To directly investigate whether HR was defective in LMS cell lines, we used a nuclease-induced genome engineered reporter system (Traffic Light Reporter (TLR) assay) to monitor HR and non-homologous end joining (NHEJ) activity in response to DNA damage[35,36]. In brief, if double-strand break DNA repair occurs via an HR-dependent mechanism in this assay, a GFP open reading frame is restored. Conversely, if NHEJ is the operative DSB repair mechanism, a frameshift places a mCherry coding sequence in-frame. In LMS cells, there was a marked increase in mCherry signal in response to DNA damage, similar to the RPEΔp53ΔBRCA1 HR-deficient control (Fig. 2d, e). Given the near-universality of LMS cell line responsiveness to single agent talazoparib, a PARP trapper, and to WEE1 and CHEK1 inhibitors, these DDRis are promising therapeutic agents for LMS patients.

**Genomic mutations highlight early evolutionary divergence of primary and metastatic LMS.** We next asked whether an LMS tumor's subtype is fixed or whether it can change at relapse. In the five primary-distant metastatic relapse tumor pairs with RNA data, all relapses maintained similar gene expression programs to the matched primary tumors. In general, all tumors from the same patient (whether occurring at different locations or time points) maintained their transcriptional subtype (see Supplementary Fig. 4A). This indicates that the LMS subtype is established early during tumor evolution and retained thereafter––even for tumors that arise from different organs separated by many years.

In addition to maintaining their defining molecular subtype, multiple samples from the same patient had shared early missense mutations in recurrent drivers (such as TP53) (Supplementary Fig. 16). To time the remaining mutations across the full cohort (70 genomes), we took advantage of the fact that many LMS undergo whole-genome duplication (WGD), thus providing a common temporal landmark to which we could compare. We detected WGD in 31/70 (44%) LMS whole-genome sequences. Using these WGD cases, the principles from Jolly and Van Loo[37] and MutationTimeR[38], a tool used to time somatic mutations relative to clonal and subclonal copy number states (see Methods), we found that only 37% of substitutions arise

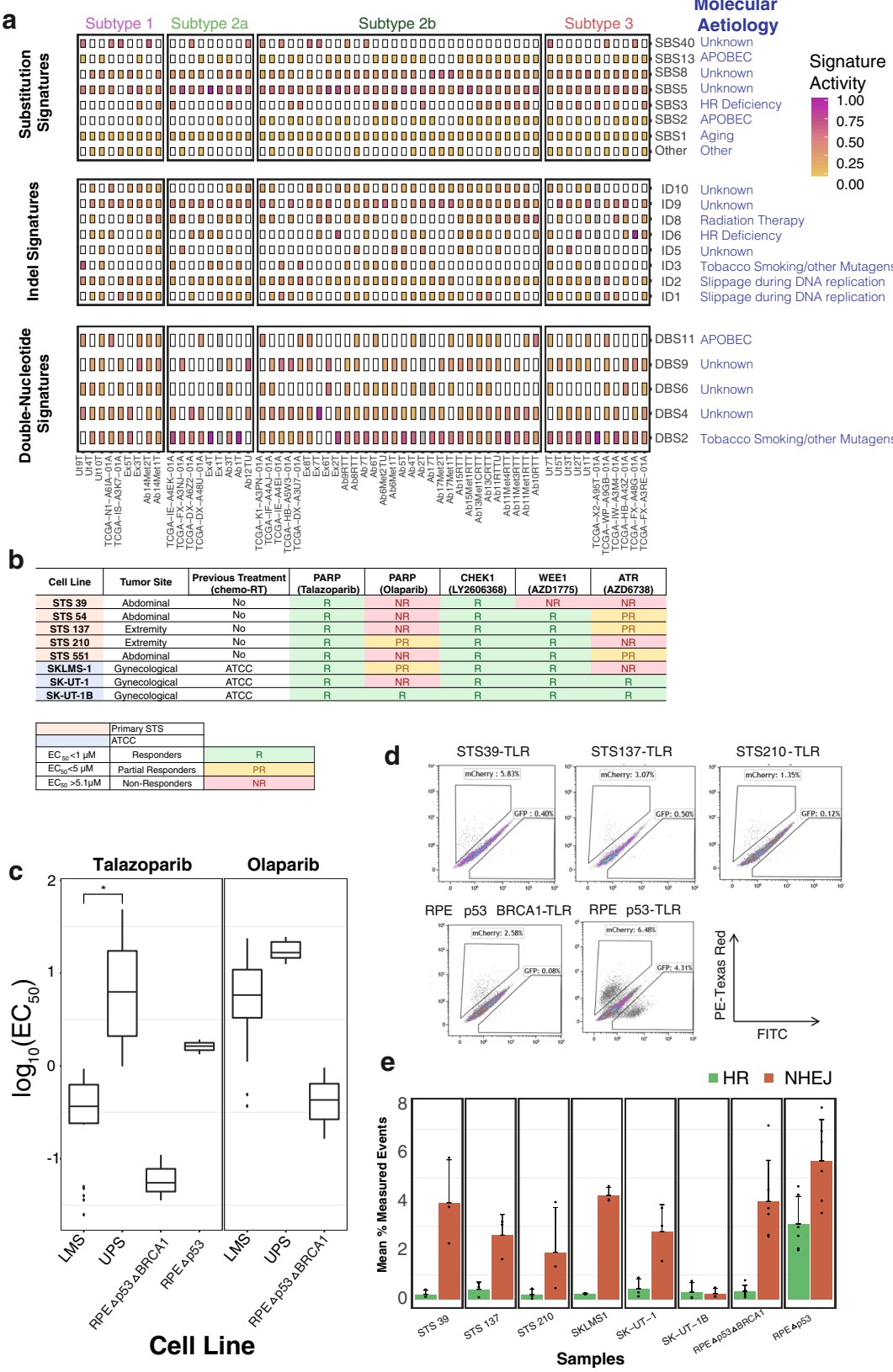

before WGD, whereas the remaining are late events (Supplementary Fig. 17). Similarly, we found that putative driver deletions preceded WGD, including loss-of-heterozygosity (LOH), copy-neutral LOH, copy-gain LOH events and homozygous deletions. These early arm-level or whole-chromosome deletions involved well-known tumor suppressor genes, such as

*RB1* (93%), *BRCA2* (87%), *CDH1* (67%), *FANCA* (60%), *TP53* (87%), *BRCA1* (42%), and *PTEN* (71%) (Supplementary Fig. 18A, B, Supplementary Data 4). Therefore, in addition to somatic *TP53* substitutions, which is likely the first alteration in the genome (Supplementary Fig. 17A, C), tumor suppressor chromosomal losses are early and initiating events in

**Fig. 2 Genomic mutation signatures in LMS and functional evaluation of defects in the DNA damage response. a** Non-negative matrix factorization (NMF)-extracted and decomposed single-substitution (SBS), indel (ID) and double-nucleotide signatures (DBS) are illustrated in the heatmaps. Common substitution signatures include SBS1, SBS5, SBS8, and SBS40. SBS3 and ID6 (HR-deficiency) are found in 64% of samples. SBS2, SBS13, and DBS11 reflect localized hypermutation events, also called 'kataegis'. ID8 represents a radiation signature, commonly seen in patients treated with radiation therapy. 'Other' substitution signatures, present in less than 5% of samples, can be found in Supplementary Fig. 13. Color refers to signature activity. **b** Evaluation of sensitivity to DNA damage response pathway, including PARPi, in soft tissue (ST) LMS cell lines (STS39, STS54, STS137, STS210, and STS551) and gynecological LMS cell lines (SKLMS-1, SK-UT-1 and SK-UT-1B). **c** Representative boxplots of EC$_{50}$ from LMS ($n = 8$) and UPS ($n = 5$) cell lines treated with the PARP inhibitors, talazoparib, and olaparib. The boxes represent the 25th and 75th percentile (bottom and top of box), and median value (horizontal band). The whiskers indicate the variability outside the upper and lower quartiles. For olaparib treatment, boxplots were generated for seven LMS and three UPS cell lines only, as growth suppression failed to occur in the remaining one LMS and two UPS cell lines along with the RPEΔp53 control. In contrast, all LMS cell lines are responsive to talazoparib (median EC$_{50}$ 0.37 μM) compared to UPS cell lines (median EC$_{50}$ 6.26 μM, $p = 0.072$, one-sided Welch's $t$-test). Detailed information for all patient derived cell lines (LMS and UPS) can be found in Supplementary Data 8 and 9. **d** The Traffic Light Reporter (TLR) assay uses a fluorescent-based system (GFP and mCherry) to determine Homologous Recombination (HR) and Non-homologous End Joining (NHEJ) efficiencies, upon induction of a double-strand break (DSB). Stable LMS-TLR (STS39-TLR, STS137-TLR, and STS210-TLR) and control cell lines (RPEΔp53-TLR and RPEΔp53ΔBRCA1-TLR) containing a single copy of the TLR I-SceI target site were generated. An I-SceI tagged with BFP was introduced to evaluate repair efficiencies. Repair of the DSB by HR generates distinct fluorescent signals (GFP$^+$), compared to NHEJ (mCherry$^+$). LMS cell lines demonstrate HR-deficiency comparable with the RPEΔp53ΔBRCA1 control cell line. In contrast, GFP$^+$ cells were detected in the HR proficient RPEΔp53 cell line. **e** The bar plot illustrates quantification of GFP to mCherry signal in each LMS cell line and controls. Intact HR (GFP$^+$) is 6X higher in RPEΔp53, compared to LMS cell lines or the RPEΔp53ΔBRCA1 control. Data are derived from eight cell lines examined over three independent experiments and the error bars represent the standard deviation. Source data are provided as a Source Data file.

leiomyosarcomagenesis, which precede DNA amplifications (Supplementary Fig. 18D).

After having established their subtypes and acquired canonical mutations, subclones within LMS tumors begin to diverge. This was particularly apparent when considering somatic structural rearrangements, a mutation type that is missed by exome or targeted sequencing. While >60% of clonal substitutions and indel mutations were shared between primary and metastatic tumor pairs, only 20% of structural rearrangements were common, suggesting the majority of these events occur after divergence (Fig. 3a, Supplementary Figs. 19A). The high frequency of clonal mutations unique to primary or metastatic tumors indicated that they evolved in parallel. This was further supported by the presence of regional hotspots of mutations, also known as kataegis[33], present distinctly in either primary or metastatic tumors (Fig. 3b). First described in breast cancer[33], this pattern of clustered substitutions has not yet been reported in LMS (Supplementary Data 5). Furthermore, one tumor (Ab17) also exhibited separate chromothriptic events between primary and both metastatic relapses (Supplementary Fig. 19B, C). Together, these data indicate that, after having acquired key drivers, LMS tumors undergo a late burst of additional changes in the form of WGD, kataegis, and/or chromothripsis. These events occur separately between primary and metastatic pairs, suggesting continued genetic diversification of both the primary and relapse after seeding of the metastases.

**Metastatic LMS branches from the primary tumor decades prior to diagnosis.** We then examined the absolute timing of mutations in LMS, both within tumors for which multiple regions have been sampled, and between paired primary-relapse tumor pairs. Multiregion sequencing on three tumors from three patients was performed (two primaries and one metastatic relapse), in addition to bulk sequencing of other tumors from the same patients (Figs. 3c and 4). We reconstructed the phylogeny of each patient's cancer using an established method (Treeomics[39]), based on the principles that shared mutations are early and private mutations are late. We validated shared or private mutations using our targeted deep sequencing panel, designed to sequence >75% of point mutations, all non-synonymous indels, and structural variants with breakpoints within known cancer genes (Supplementary Fig. 20). In addition, we corroborated our phylogenies and identified subclonal populations within LMS tumors

by using a Dirichlet process to model cellular fractions of variants, as previously described[40,41] (See Methods, Supplementary Figs. 21–23). In all patients, *TP53* mutations and chromosomal losses of chr 10 (*PTEN*), 13 (*RB1*) and 17 (*TP53*) were part of the common trunk of the phylogenetic tree, thus supporting our bulk, single tumor sequencing data that these are the earliest events in the genome. We also noted alterations in the Wnt/β-catenin signaling pathway that were part of the trunk of our phylogenies (Fig. 3a and Fig. 4). Strikingly, we found that metastatic relapses branched off early from the primary tumor in both multiregion cases (Fig. 4). This finding was also supported in the (bulk) primary-metastasis pairs. In patient Ab6, the first metastatic relapse (MR1) appears to have branched off before the primary (Dx) and second metastatic relapse (MR2) separate. These data support a model of branching evolution where clones diverge from a common ancestor and evolve in parallel resulting in multiple clonal lineages[42].

To time the divergence of primary and metastatic tumors we established the rate of mutations in LMS and its subtypes (expressed as substitutions per gigabase per year; see Methods and Anderson et al.[9]). Signature 1 mutations (SBS1) are proportional to chronological age and so can approximate the number of mitoses a cell has undergone during the lineage of cell divisions since the fertilized egg[43] (Supplementary Fig. 24). Using SBS1 as a molecular stopwatch, we determined that the primary and relapsed LMS cancers can be estimated to have diverged from one another 10–30 years prior to diagnosis––providing many years to accumulate the private mutations we observed (Fig. 3a). The divergence of multiple metastases from each other also occurred in the patients' distant past, over a similar time period (10–30 years pre-diagnosis). This leads to the question: how long ago did adjacent tumor foci or adjacent tumor regions––which are separated only by a few centimeters––diverge from one another? As expected, regions at an increased spatial distance were also more genetically disparate. This is particularly evident in patient Ab6, from whom we sequenced five regions from two tumor foci from their second metastasis (three regions from focus 1 and two regions from focus 2), in addition to both their primary tumor and first metastasis (Fig. 4a). Taken together, from earliest to latest divergence: ~30 years prior to the initial diagnosis of a large abdominal LMS, Ab6's primary tumor and first metastatic relapse (MR1) diverged; five years later (~25 years prior to initial diagnosis), the ancestor of metastatic relapse 2 (MR2) then

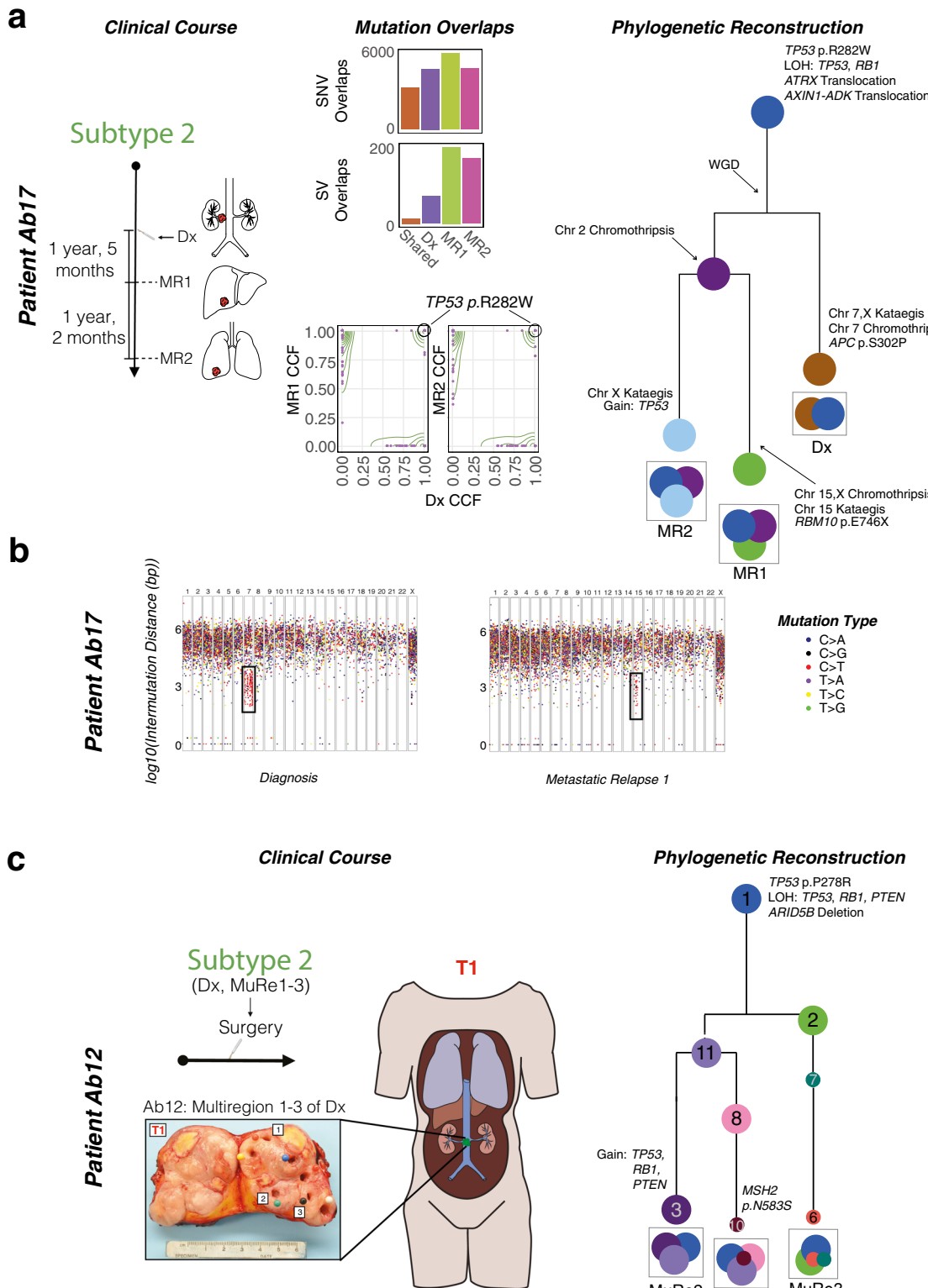

diverged. After having established itself in the patient's vastus lateralis, MR2 diverged again ~6–7 years pre-diagnosis leading to two independent foci located only ~1.5 cm from each other; and then, within each parallel evolving focus, further genetic diversification led to heterogeneous tumor regions separated by <1 cm apart from one another, which diverged ~2–4 years after initial diagnosis. A similar analysis in patient Ab11 confirmed the early origins of LMS and the early divergence of its lethal relapse

(Fig. 4b). Thus, both multiregion and paired primary-metastatic relapse sequencing establish the early and parallel genomic evolution of LMS cancers.

## Discussion

By taking a broad view of LMS using both whole-genome and RNA sequencing of primary, metastatic and multi-site sampled

**Fig. 3 Clonal evolution and phylogenetic analysis of LMS tumors. a** The clinical course of patient Ab17 with a primary (Dx) and two metastatic relapses (MR1 and MR2) is shown (far left, n = 3 samples). Structural variant (SV) overlaps (middle, top) and the cancer cell fraction (CCF) of single-nucleotide variants (SNVs) (middle, bottom) illustrate that there are many SVs and clonal variants that arise independently in the primary tumors and metastatic relapses. Phylogenetic reconstruction of Ab17's tumors can be seen on the far right. The founder clone harbors a pathogenic *TP53* substitution, whole-genome duplication (WGD), as well as loss-of-heterozygosity (LOH) events encompassing *TP53* and *RB1*. The color of each circle represents a distinct clone population. The clonal trajectory and final composition are shown per sample. Branch lengths are proportional to Treeomics mutation assignments. **b** Rainfall plots of patient Ab17 in diagnosis and their first metastatic relapse illustrate differential kataegis events at different chromosomes between the two time points. Targeted sequencing data were used to confirm kataegis events were unique to each specimen. **c** The clinical course for patient Ab12 is depicted, which involved no prior treatment and only surgery. The primary specimen at diagnosis (Dx) was located in the inferior vena cava. The tumor was bisected and punch-hole biopsied in three physically distant multiregion (MuRe) locations (n = 3 samples). The phylogenetic reconstruction of this tumor is shown on the right of the schematic and a photo of the resection. The founder clone harbors a pathogenic *TP53* substitution, as well as LOH events encompassing *TP53*, *RB1*, and *PTEN*. Larger circles represent major clones, whereas smaller circles represent subclones. The color of each circle represents a distinct clone population. The clonal trajectory and final composition are shown per sample. Branch lengths are proportional to Treeomics mutation assignments, except for clones 8,10 and 6,7 where DPClust mutation assignments were used to stratify the sample.

tumors, we identified the genomic basis of molecular subtypes, found clinically relevant mutational signatures, and outlined the disease's evolutionary trajectory.

The overarching theme from this study is that the seeds of a LMS cancer's aggressiveness are planted very early. The tumor's originating tissue type, either digestive, vascular, and/or gyneco-logical smooth muscle, combined with its degree of dedifferentiation, is one of the primary determinants of LMS molecular subtypes and the patient's ultimate survival. Overall, the three subtypes we describe bear similarity to the transcriptomic subgroups previously described[11,13], although a meta-analysis is warranted to assess the effect of different NGS platforms. Thus, there is a framework emerging from LMS multiomic efforts and ongoing collaborative initiatives that will generate consensus molecular definitions of LMS subtypes, including the resolution of nomenclature, clinical information, and biomarkers that will aid patient stratification and/or prognostication.

In this study, we describe two LMS subtypes with poorer survival, subtypes 1 (dedifferentiated) and 3 (primarily gynecological), that are also defined by a prevalence in somatic dystrophin deletions and high immune infiltration. Consistent with this, dystrophin deletions are also found in non-myogenic cancers at higher or similar frequencies to other well-known tumor suppressor genes and associated with significantly poorer overall survival[44], thus highlighting dystrophin's role as an emerging tumor suppressor. Importantly, these subtypes have the potential to help differentiate the prognosis of tumors from the same anatomic site. Implementing LMS molecular subtyping, including matching tumors to normal smooth muscle tissue, can be used to reveal its originating tissue at diagnosis. Specifically, an LMS tumor in our cohort resected from an extremity site, clustered with both the subtype 1 group and normal gynecologic tissue, indicating it was in fact a uterine metastasis, which correlated directly with the patient's clinical history. The ability to define the origins of this "extremity" tumor may provide different, more effective therapeutic options for this patient. This approach could also aid in resolving discrepancies when the previous pathology was inconclusive and unable to document a primary site of LMS. This is, for example, an ongoing conundrum with the diagnosis of uterine leiomyomas[45].

The molecular subtypes of LMS, as defined by the originating tissue types and dedifferentiation, then determine the overall burden of secondary mutations that are acquired (across all variant classes)––with the most dedifferentiated tumors and gynecological LMS acquiring the highest burden. Once the cancer is clinically diagnosed, our data suggest that many patients will already have another tumor growing in parallel. Again, one sees that the seeds of LMS are established much earlier––in LMS,

lethal metastatic disease can arise up to three decades prior to clinical diagnosis, consistent with other adult cancers[38]. The exact timing of primary-relapse divergence requires validation in larger cohorts of LMS that include all molecular subtypes and anatomic sites.

With its molecularly defined background established, additional genetic events are layered on top of the LMS tumor genome. Early mutations in *TP53* appear to be near-universal in LMS. The LMS founder clone is also characterized by arm-level or whole-chromosome LOH events, most frequently involving the *RB1* and *TP53* genes. Other patient-specific alterations also arise early, including *ATRX* deletions and Wnt/β-catenin alterations. From then, mid-late events account for the majority of LMS mutations, contributing to the tumors characteristic karyotypic instability––including dramatic genome-wide duplications, kataegis, and chromothripsis. These events generate widespread genetic diversity, as indicated by the presence of multiple subclonal and clonal populations within a single primary or relapse LMS tumor.

Regardless of molecular subtype, more effective therapies are urgently required in LMS, especially in patients with advanced disease. We show and validate that simple substitution combined with indel signatures reflect HR-deficiency in LMS, which can be exploited therapeutically in the form of PARP trappers and DNA damage response inhibition therapy. Together, HR-deficiency signatures were present in 64% of LMS, lower than previously reported by exomes[5]. Furthermore, we demonstrate that LMS cell lines are highly responsive to talazoparib treatment, in contrast to other sarcoma and HR proficient cell lines. Here, we use stricter sensitivity thresholds to previous reports, thus further refining the sensitivity of LMS cell lines to PARPi[5]. Additionally, LMS cell lines are responsive to DNA damaging agents CHK1 and WEE1. The present study suggests that LMS patients should be considered for comprehensive HR profiling for use of mutational signatures as biomarkers of therapy.

To conclude, LMS, the catch all-label for any primary malignant smooth muscle neoplasm, encompasses at least three distinct entities. We report that the utility of DNA damage response inhibitors is promising and warrants further exploration. Finally, the molecular evolution and timing of LMS provide a broad window for early intervention and risk stratification for these patients.

## Methods

**Sample acquisition and patient characteristics.** Leiomyosarcoma (LMS) tumor and matched-blood and/or adjacent normal tissue samples were collected at Mount Sinai Hospital and the University Health Network (UHN) in Toronto, Canada in accordance with each institutions' Research Ethical Board (REB) guidelines. Written informed consent was obtained from all patients enrolled in the LMS

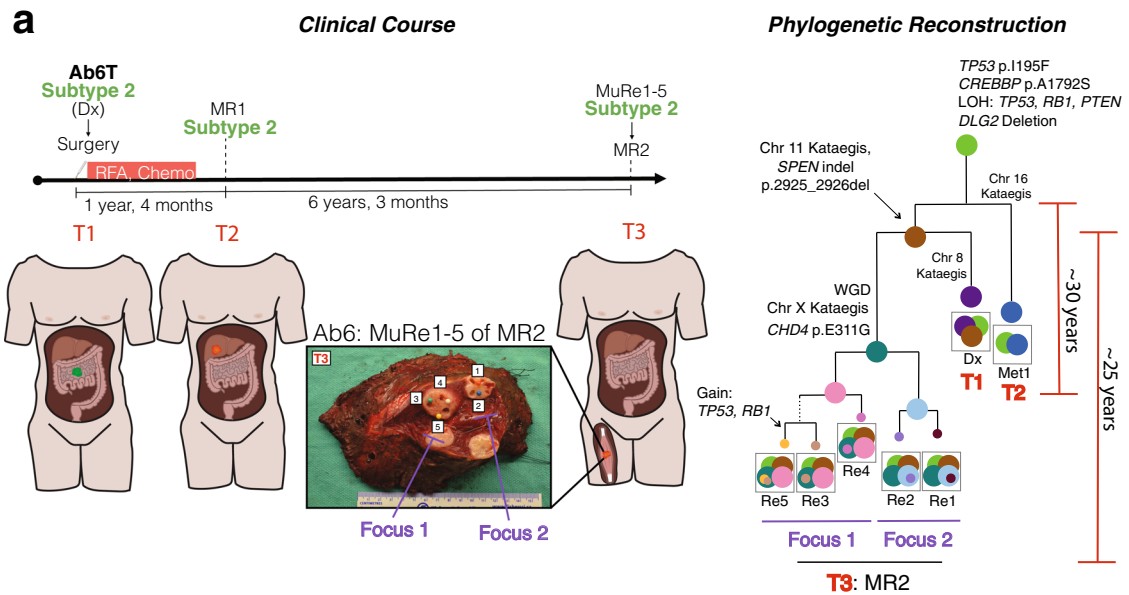

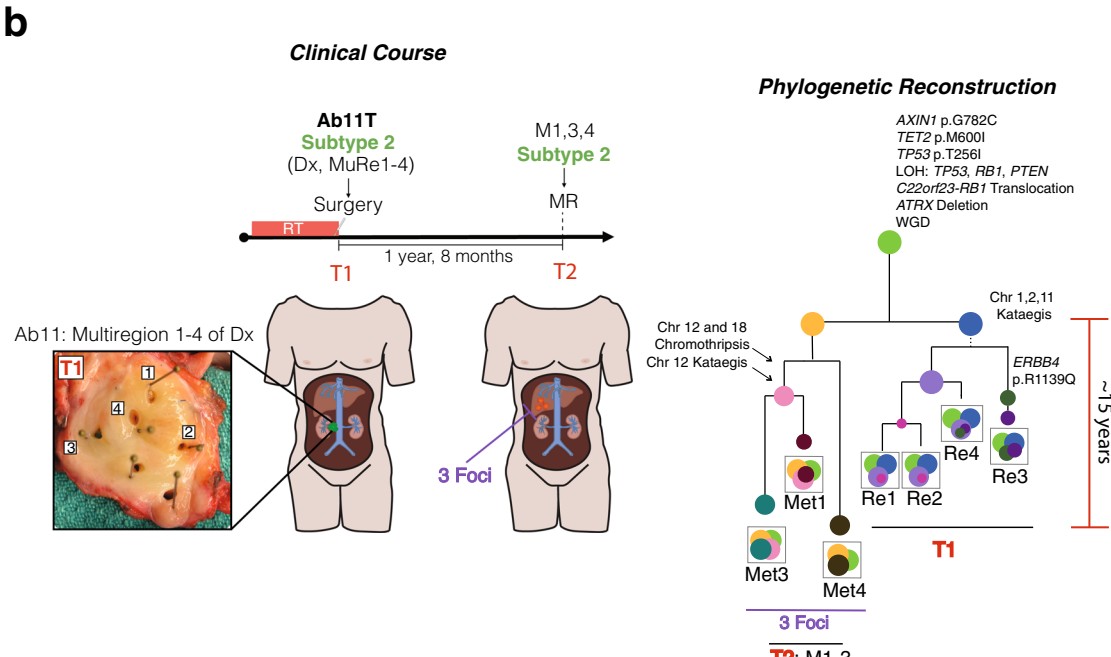

Genomics Program, including permission to publish indirect identifiers. Each specimen underwent extensive pathological review by expert pathologists. LMS was required to have unequivocal histologic or immunophenotypic evidence of smooth muscle differentiation. Detailed clinical information (age at presentation, sex, tumor site, stage, etc.) was obtained (Supplementary Data 10). We note a slight bias towards grade II and III LMS in the Toronto cohort, which includes 29 untreated tumors (24 primary tumors, five metastatic relapses) and 13 tumors treated with radiation (seven primaries, six metastatic relapses). Overall, the patients' clinical features and demographics were typical of LMS: the average age at diagnosis was 56.5 years (28–87 yrs.); the male to female ratio was 1:2.6 (abdominal/ extremity only).

For multiregion samples, tumors were bisected en face and tumor regions (or sites) were biopsied, pinned, and numbered. Biopsy punches were 4–8 mm in diameter and sliced longitudinally. Half of the biopsy was processed for histology and quality control, while the other was used for DNA/RNA extraction using a Qiagen Allprep Micro Kit. Photos were taken after sampling to document the relative location of each punch biopsy. H&E staining was used to estimate the viability of each sample.

**LMS samples from the cancer genome atlas (TCGA)**. LMS samples that were sequenced as part of TCGA were downloaded from the NIH Genomic Data Commons (GDC) Data Portal [http://gdc.nci.nih.gov/]. Six-panel pathologists reviewed LMS cancers as part of the TCGA SARC program[6]. We removed any sample which failed TCGA SARC review, even if the sample is available for download on the GDC. Thus, 18 tumor/normal whole-genome and 80 transcriptome sequences (known as the validated TCGA SARC cohort) were downloaded from the NCI GDC and processed with the Toronto cohort. One genome and one transcriptome were removed due to the detection of a pathogenic KIT variant (TCGA-MO-A47R-01A).

**DNA/RNA extractions, whole-genome and RNA sequencing of LMS patients**. DNA and RNA were extracted from fresh-frozen LMS material or patient-derived cell lines. For fresh frozen tumor material, DNA from matched-blood and/or fresh-frozen matched normal tissues was used as a normal reference. RNA libraries were constructed using the NEB Ultra II Directional mRNA kit. The resultant libraries were checked on a Bioanalyzer (Agilent) and quantified. RNA starting material for each sample in the Toronto cohort is available in Supplementary Data 11. Libraries

**Fig. 4 Parallel evolution of LMS tumors.** The clinical courses of two patients with LMS are shown (samples per patient >3). For the phylogenies, larger circles represent major clones, whereas smaller circles represent subclones. The color of each clone represents a distinct clone population. The clonal trajectory and final composition are shown per sample. Branch lengths are proportional to Treeomics mutation assignments. (**a**) Patient Ab6 was treated with radiofrequency ablation (RFA) and chemotherapy (chemo). They had three tumors at three separate time points (T1, T2, and T3). The primary tumor at diagnosis (Dx) was located in the small intestine, while the first metastatic relapse (MR1) was located in the liver. The second metastatic relapse (MR2) was multifocal and detected in the vastus lateralis (thigh muscle). MR2 was bisected and biopsies were taken from five distinct sections from both foci (Regions/Re 1–5). Following bulk (Dx, MR1) and multiregion (MR2) sequencing, phylogenetic reconstruction can be seen on the right. Early substitutions in *TP53*, *RB1* and *CREBBP*, as well as LOH events of chromosomes 10 (*PTEN*), 13 (*RB1*), and 17 (*TP53*) are observed in the founder clone of this patient's tumors. Chromosome 11 kataegis events and an *SPEN* deletion were common to Dx and MR2, but not MR1. Genome doubling and chromosome X kataegis occurred only in MR2. Metastatic multifocal nodes greatly resemble each other. Using clock-like mutagenesis, the Dx and MR1 diverge approximately 30 years pre-diagnosis in these patients, while Dx and MR2 diverge approximately 25 years pre-diagnosis. (**b**) Patient Ab11 was treated with radiation therapy (RT) and had two tumors at two separate time points (T1 and T2). The Dx was located at the posterior aspect of the right kidney and inferior vena cava. This tumor was bisected en face and biopsies were taken from four regions. The metastatic relapses (MR1-3) were taken from the liver. Following bulk (T2: MR1-3) and multiregion (T1: Dx, Regions/Re1-4) sequencing, phylogenetic reconstruction can be seen on the right. Much like Ab6, early losses of chromosomes 10, 13, and 17 are observed. Also seen early are *AXIN1* and *TET2* point mutations, an *ATRX* deletion, and a *RB1* translocation. Chromosome 2 kataegis events are unique to Dx, while chr 6 chromothripsis events are unique to MR1-3. A chromosome12 kataegis event occurs only in 2/3 liver metastases. Using clock-like mutagenesis, Dx and MR1-3 diverge approximately 15 years pre-diagnosis in this patient.

were then multiplexed, clustered, and sequenced on a HiSeq2500 in high throughput mode. WGS library was constructed using TruSeq DNA PCR-free kit from Illumina. For genome sequences, paired-end FASTQ files were aligned to the human genome (hg19/GRCh37) using BWA-MEM[46] (v.0.7.8). Picard MarkDuplicates (v.1.108) was used to mark PCR duplicates. Indel realignment and base quality scores were recalibrated using the Genome Analysis Toolkit[47] (v.2.8.1).

**Gene expression and clustering of LMS.** Gene expression was performed on RNA-Sequencing data from 51 LMS samples sequenced in Toronto and 79 samples from TCGA. Samples from all sources were analyzed with a common gene expression pipeline, from raw fastq data to expression counts, to prevent the addition of in silico noise. RNA-seq reads were aligned on human genome using STAR[48] (v.2.4.2), while HT-Seq was used to count reads aligning on every single gene, after removing PCR duplicates and reads mapping on ribosomal RNA, miRNA, and small nucleolar RNA. Normalization was performed using Trimmed Mean of *M* value (TMM) method in EdgeR[49] on genes with at least one reads per million base in at least 3 samples. Normalized data were then log-transformed and the removeBatchEffects function from the limma package in R was used to remove any source of variability from different experiments[50]. Unsupervised Principal Component Analysis (PCA) was done using R (v. 3.4.4). Unsupervised clustering was generated using a hierarchical algorithm based on the average-linkage method. Only genes displaying a variance greater than 0.9 in the expression level across the whole panel were chosen to generate the hierarchical clustering. The distance between two individual samples was calculated by Pearson distance with the normalized expression values. K-mean clustering algorithm with 100 iterations was used to select different LMS subgroups.

Validation cluster was performed using UMAP[15] and DBSCAN[16]. UMAP, like t-SNE[51], is a non-linear dimensionality reduction method, while PCA can only capture linear correlations. We applied a step of low variance genes filtering before each dimensionality reduction, which helped reduce dataset differences in the most evident cases. We first obtained gene expression counts (TPM) by running samples through the toil-RNASeq pipeline[52] (which includes cutadapt v.1.9, STAR v.2.4.2a, and RSEM v.1.2.25). The results were then log2 normalized and clustered together with a reference tumor dataset obtained from the UCSC Treehouse Childhood Cancer Initiative (https://treehousegenomics.soe.ucsc.edu/explore-our-data/, Compendium v.9). In total, 12, 419 RNA-seq samples were amalgamated by this initiative, including datasets from TCGA and St. Jude's Children's Hospital. We repeated this process independently for 1735 normal tissue samples obtained from the GTEx Consortium database[17,18]. When comparing the tumor samples with the normals, we trained the dimensionality reduction map on normals only and projected the LMS cohort on the result. This allowed us to match each cancer to the closest normal tissue and remove batch effects arising from the inherent differences between malignant and normal tissues.

**Detection of gene fusions.** We detected gene fusions using a custom tool that integrates multiple independent fusion algorithms: Defuse[53] (v.0.6.2), ChimeraScan[54] (v.0.4.5), STAR[48] Fusion (v.0.7.0), MapSplice (v.2.1.9), and FusionCatcher[55] v.(0/99.4d_beta(72-76)) (Fuligni et al., Under preparation). The tool then performs a dynamic realignment and removes artifacts found in normal tissue. A total of 1277 normal (non-neoplastic) samples from 43 different tissues were obtained from the NHGRI GTEx consortium (database version 4) and used to remove these artifacts. Putative fusions were validated by de novo assembly.

**LMS survival analysis.** Clinical information and complete follow-up were available for all 113 patient cases. Specifically, overall survival (OS) and disease-specific survival

(DSS) were calculated from the time of diagnosis to death or last follow-up. For OS the event is the death of any cause whereas with DSS the event is death from LMS only. TCGA survival data (OS and DSS) was obtained from Liu et al.[56]. Survival data were analyzed using the Kaplan–Meier and log-rank Mantle–Cox methods. The limit of significance for all analyses was defined as having a $P$ value < 0.05.

**Somatic variant calling.** We detected somatic mutations using established tools (MuTect2[57] (part of GATK v.3.8) and Delly[58] v.0.7.1), and used custom filters as previously described[9]. Due to the presence of small deletion artifacts in one LMS sample, all small deletions (<500 bp) with less than 6 read pair support were filtered out if they did not have additional split read support. Depth-based and allele-specific copy number were detected from whole-genome sequences using BIC-seq[59] (v.1.2.1) and Battenberg[40] (v.3.2.2), respectively. A copy number loss or gain was determined if (1) the log2 ratio by BIC-seq was below −1 or above 0.58 and (2) allele-specific copy number by Battenberg indicated a total copy number >2 for gains and <2 for losses. Any discrepancies in copy number calling were resolved by visual inspection.

**Immune infiltration analysis.** Leukocyte content of TCGA LMS samples was obtained from Thorsson et al.[24] (Supplementary Table 2 from Thorsson et al.). A Mann–Whitney $U$ test was run to confirm a statistically significant difference ($p$ value = 2.769e–05) of leukocyte fraction between subtype 1 and subtypes 2/3 cancers.

**Mutation signature extractions and analysis.** Variants from whole-genome sequences were used for mutation signature analysis. Multiregion LMS cancers were merged and ran as a single sample. For signature analysis, a de novo extraction was performed on the catalogue of LMS point mutations to produce consensus mutational signatures. These signatures were deciphered using a previously described, recently updated computational framework that optimally explains the proportion of each mutation type found in the catalogue and then estimates the contribution of each signature to the mutation catalogue (SigProfiler v.1.0.9). In addition to simple substitution signatures, this updated framework additionally detects indel signatures (ID) and double-substitution signatures (DBS). Extracted signatures are then compared true consensus mutational signatures to the previously curated COSMIC list and quantified their similarity using cosine similarity as previously done[43]. We report >0.9 cosine similarity between LMS signatures and the COSMIC list. SBS3 is notoriously difficult to discern from other "flat" signatures such as SBS5 or SBS8[7]. As a primary validation, we obtained the ground-truth simulated SBS3 mutations from 1000 samples from the ICGC/TCGA PCAWG initiative (https://www.synapse.org/#!Synapse:syn18497223)[7]. Concordance was compared between simulated data and results obtained from our mutation signature pipeline. As a secondary validation, we noted a decrease in cosine similarity below 95% when SBS3 was not included in the decomposition.

**Culturing of LMS and UPS Cell Lines.** Primary patient-derived LMS cell lines (STS39, STS54, STS137, STS210, STS551) and UPS cell lines (STS148, STS162, STS235, STS309, STS548) were established from pathologically reviewed surgical specimens in accordance with institutional research ethics as described elsewhere[60]. SKLMS-1, SKUT1, and SKUT-1B and Hs 789.Sk cell lines were obtained from ATCC (Manassas, VA, USA). RPEΔp53 and RPEΔp53ΔBRCA1 were kind gift of D. Durocher. All cell lines are routinely authenticated by STR-analysis at The Center for Applied Genomics (TCAG), SickKids in Toronto, and tested negative for mycoplasma (ABM, Canada). All cases were reviewed by a dedicated sarcoma pathologist (BCD) (See Supplementary Data 8 and 9). LMS cell lines maintained

smooth muscle expression (Supplementary Fig. 25). STS39, STS54, STS137, STS210, STS551, STS148, STS162, STS235, STS309, STS548, and SKLMS1 cells were cultured in GlutaMAX-supplemented DMEM/F12 (Gibco, Thermo Fisher Scientific, Waltham, MA, USA) +1% Penicillin/Streptomycin (Thermo Fisher Scientific) and 10% heat-inactivated fetal calf serum (FBS) (Wisent, St-Bruno, Canada) at 37 °C, 5% CO2. SKUT1, SKUT-1B, RPEΔp53 and RPEΔp53/ΔBRCA1 were cultured in GlutaMAX-supplemented DMEM (Gibco, Thermo Fisher Scientific, Waltham, MA, USA) +1% Penicillin / Streptomycin and 10% FBS at 37 °C, 5% CO2.

**Traffic light reporter (TLR) assay**. For the quantitative detection of HDR and NHEJ events, a TLR assay was used. To establish LMS-TLR cell lines, cells were infected with pCVL.TrafficLightReporter. Ef1a.Puro lentivirus at a low multiplicity of infection (MOI 0.3–0.5) and selected with puromycin (15 μg/μl). $7 \times 10^5$ cells were nucleofected with 5 μg of pCVL.SFFV.d14GFP.Ef1a.HA.NLS.Sce(opt). T2A. TagBFP plasmid DNA in 100 μL of electroporation buffer (25 mM Na2HPO4 pH 7.75, 2.5 mM KCl, 11 mM MgCl2), using program T23 on a Nucleofector 2b (Lonza). After 96 h, GFP and mCherry fluorescence were assessed in BFP-positive cells using a Gallios (Beckman Coulter, USA) flow cytometer (gating strategy described in Supplementary Fig. 26). Analysis was performed using Kaluza Analysis software (Beckman Coulter, USA).

**DNA damage response inhibition (DDRi) assays**. LMS and UPS cells were seeded at the SMART Facility (LTRI) in 384 well plates. Plates were incubated for 4–6 h to ensure cell attachment, and then were sprayed with 24 concentrations (0.013–10 μM) of DDRi compounds (Drug Discovery, OICR). After 7 days, viability was measured with ATPlite (Perkin Elmer, US). Plate maps were scrambled and drugs were not sprayed on edges to reduce variability. Data were normalized to cells treated with vehicle (DMSO) and represented as a percentage. The following drugs were used in the course of this study: olaparib (SelleckChem, Houston, TX, USA, or Astra Zeneca, Cambridge, UK), talazoparib (SelleckChem), selective ATR inhibitor (AZD6738), CHK1 inhibitor (LY2606368), WEE1 inhibitor (AZD1775). Concentration and duration of treatment are indicated in the corresponding figure legends. 2/5 UPS, 1/8 LMS, and the control RPEΔp53 cell line did not show any growth suppression with olaparib treatment. With 3/5 UPS and 7/8 LMS cell lines we were able to calculate $EC_{50}$ values (GraphPad Prism 8) and this quantitative data was used to generate the box plot shown in Fig. 2c.

**Phylogenetic analysis**. To investigate mutational heterogeneity and evolutionary dynamics in LMS, we used Treeomics[39], DPClust[40], and MutationTimeR[38]. The cancer cell fractions were calculated using formulas from McGranahan et al.[61]. First, for our tree construction, we used Treeomics which also annotated trees with 'likely' driver substitutions. Phylogenetic trees were further manually annotated with other alterations (indels, SVs and CNVs) of known LMS driver genes, chromothripsis, and kataegis events. Treeomics calculates coverage, purity, and VAFs for all variants and determines reliability scores by combining evidence for each possible mutation pattern across all variants and samples. It uses a Bayesian inference model to determine the posterior probability of whether a variant was present or not in each sequenced sample. Variants shared between two samples in the same patients are early, clonal mutations that generate the trunk of the phylogenetic tree. Conversely, variants unique to a particular sample are later events and populate the branches. We further validated shared and unique variants in samples by creating a union of all variants in patients with multiple samples (multiregion or paired primary-relapse samples) and genotyped each position in each 30X bam file and the targeted (~700X) data for select cases, ensuring a minimum mapping quality of 35 and a base quality of 20.

To determine the subclonal architecture of mutations in LMS, we applied DPClust to multiregion and paired primary-relapse samples in patients with at least three samples. DPClust is based on a Dirichlet Process that clusters variants of a similar cancer cell fraction and estimates the number and prevalence of cancer cell populations (https://github.com/Wedge-lab/dpclust). DPClust yields a series of clusters and mutation assignments. Clusters composed of less than 1% of the total mutations or those that were defined by mutations with low CCFs in all samples were omitted as these are consistent with artifacts. To be considered for inclusion as a circle dot in the tree, mutations must represent at least 5% CCF, 5% of the total mutations of the cluster, and have at least five mutations in that sample. Treeomics phylogenies were validated using 'DPClust tree builder,' which infers relationships between pairs of clusters using cluster CCF confidence intervals. Briefly, each cluster pair is classified as possibly fitting at the same level in the tree (when confidence intervals overlap), having an ancestral relationship (when one cluster's intervals are greater than or less than the others) or branching (when intervals are "greater than" and "less than" in different samples). The root node is selected as the cluster that has the highest summed CCF across all samples. The base tree is then constructed by placing all clusters that have exactly one possible ancestor onto the tree. To place the remaining clusters on the base tree, both the cluster pair classifications and the cluster CCFs are utilized. A variant of the pigeonhole principle is then applied, which states that the sum of CCFs of all tree nodes at a particular level cannot exceed one. The level of a node on the tree is defined by how many steps must be taken to reach the root of the tree, plus 1 (For example, the

root is level 1 and nodes directly below are level 2). As the CCFs of clusters are imprecise estimates, we further introduce a parameter leeway that is the amount of CCF the sum can go over 1 before the pigeonhole principle metric is violated. A cluster is placed on the tree when the pigeonhole principle metric is met. If a cluster fits in multiple places, then additional trees are added that represent the different fits. Only one solution was found for Ab6, Ab11, and Ab17. Two plausible solution trees are shown in Supplementary Fig. 21. Treeomics phylogenies were modified to include DClust-generated clonal nodes and endpoints.

Both DPClust and Treeomics flagged three Ab11 samples (Ab11MuRe3RTT, Ab11MuRe4RTT, and Ab11Met1RTT) for low sample purity. Phylogenetic trees were specially curated for this patient (for example, the *TP53* substitution was not detected by mutation callers but predicted to be present by treeomics. Further, WGD was not detected but predicted to be present by DPClust, see Supplementary Fig. 27). As such, only for these three samples with incorrect ploidy that would underestimate the true CCFs, clusters with CCF between 0.45 and 0.55 were merged with another cluster that had the same CCF pattern across samples and fell within the same confidence intervals. The CCF for this merged cluster was doubled to reflect the true ploidy of the samples and obtain the correct CCF (also see Dentro et al.[41]). For Ab11, the correct ploidy solution was inferred to generate DPClust validation phylogenies. Targeted resequencing was used to verify variants were unique or shared between samples in Ab11. Of note, patient Ab11, unlike other multiregion patients Ab6 or Ab12, underwent radiation therapy treatment which may have affected tumor purity at biopsy (See Supplementary Data 10).

Lastly, to time variants relative to chromosome amplifications, we used MutationTimer (https://github.com/gerstung-lab/MutationTimeR). MutationTimeR evaluates the multiplicity state of a mutation and the copy number of the segment on which the mutation resides. If the mutation is on a gained chromosome that has a multiplicity greater than one it is clonal [early], whereas if the mutation is on a gained chromosome and has a multiplicity equal to one it is clonal late. If the mutation falls in a region that has a normal diploid copy number or is a loss it will classify clonal mutations as clonal not specified. If the cluster with the highest assignment probability is subclonal, then the mutation is assigned subclonal.

**Reporting summary**. Further information on research design is available in the Nature Research Reporting Summary linked to this article.

## Data availability

Raw sequencing data generated in this study have been deposited in the European Genome-phenome Archive (EGA) with the accession numbers EGAS00001004783 (RNA-seq) and EGAS00001005341 (WGS). Published LMS samples that were sequenced as part of TCGA were downloaded from the NIH Genomic Data Commons (GDC) Data Portal [http://gdc.nci.nih.gov/]. GTeX RNA-sequencing data was downloaded from the GTeX data portal [https://www.gtexportal.org/home/]. Source data are provided with this paper.

## Code availability

Custom code described in this study is available at github.com/shlienlab. The code used in the https://github.com/shlienlab/SVetect repository was used to filter and classify structural variants. Simple point mutations and indels were filtered and annotated using the code in the following repositories: https://github.com/shlienlab/ShlienLab.Core.SSM (initial filtering), https://github.com/shlienlab/ShlienLab.Core.Filter (additional filtering), https://github.com/shlienlab/cosmic.cancer.gene.census (COSMIC annotation).

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

## Acknowledgements

We thank Brian Chow and Colin Elliot for pathologic expertise. The authors thank Sally Burtenshaw, Sarah Rachel Katz, and Anthony Griffin for clinical data. We thank the Centre for Applied Genomics (TCAG) NGS facility for their sequencing services. We thank Nadine Tan for her scientific illustrations. R.G., B.C.D., and A.S. received funding from the Panov Foundation and the Martin E. Blackstein Patient Donation Account. A.S. and R.G. received financial support from a University of Toronto McLaughlin Centre Accelerator Grant and a Canadian Institutes of Health Research (CIHR) Project Grant.

M.B., R.G., and A.S. received support from the Department of Surgical Oncology, University Health Network. Biobanking was supported by the Jim Chamberlain Sarcoma Research Fund, Princess Margaret Cancer Centre and the Rubinoff Gross Chair in Orthopaedic Oncology (JSW), Mount Sinai Hospital. N.D.A is personally supported by a Canada Graduate Scholarship and a SickKids Restracomp Award. AS is partially supported by an Early Researcher Award from the Ontario Ministry of Research and Innovation, the Canada Research Chair in Childhood Cancer Genomics, and the Robert J. Arceci Innovation Award from the St. Baldrick's Foundation.

## Author contributions

A.S. and R.G. designed the study. N.D.A., Y.B., F.F., M.L, F.C., A.M, D.T., K.Y., H.H., M.Z., and N.L. performed experiments. N.D.A. Y.B., F.F., M.L., and L.B.A. collected and analyzed data. R.G., S.G., R.V., C.W., J.J.D., I.L.A., J.W., P.F., C.J.S., M.R., M.Q.B., and D.D. contributed reagents, tissue, and clinical data. N.D.A., A.S., and R.G. wrote the manuscript. S.D., M.G., L.B.A, I.L.A., A.R.A.R., R.A., R.M., D.D., B.C.D., R.G., and A.S. provided technical support and conceptual advice. A.S. and R.G. oversaw the study. All authors have approved the manuscript.

## Competing interests

The authors declare no competing interests.
