## [Peer Review File · Nature Communications]

REVIEWER COMMENTS

Reviewer #1 (Remarks to the Author): Expert in leiomyosarcoma genomics

This is a very interesting manuscript on an in-depth analysis of leiomyosarcoma, one of the more common forms of sarcoma. The study represents the largest cohort of cases examined by gene expression profiling and whole genome sequencing. The work confirms prior studies by others that indicated that at least three distinct molecular subtypes exist in leiomyosarcoma and provides exciting new data that improves our understanding of this neoplasm.

I am quite enthusiastic about the study but have several questions:

1 Page 5, lines 21 – 23: "Full RNA sequencing analysis for expression and fusion gene analysis was also carried out for 130 (51 Toronto, 79 TCGA) transcriptomes using an informatics framework with 97.95% sensitivity and 82.78% accuracy (Fulgini F, in preparation)." – what is actually meant with "informatics framework with 97.95% sensitivity and 82.78% accuracy"? More importantly, can the authors rely on and cite an apparently new method/computational tool that is not yet published and thus is not even described in any detail?

2 I wonder whether the sentence "Specifically, an LMS tumor in our cohort resected from an extremity site, clustered with both the subtype 1 group and with normal gynecologic tissue, indicating it was in fact a uterine metastasis." is perhaps a bit of an overstatement. This suggestion would have carried more weight if there were some clinical support that would suggest that the extremity tumor did indeed originate in the uterus.

3 The cell lines used in this study are very poorly described; the reference provided (J Transl Med 14, 67 (2016)) only mentions one of the 5 cell lines that were newly derived in Toronto. Data should be provided to indicate that these cell lines have maintained smooth muscle differentiation. Regarding the MES-SA cell line – why is this cell line included in this study as an LMS cell line? In ATCC it is described as a uterine sarcoma. In the original publication the tissue of origin was described as "poorly differentiated sarcoma, consistent with elements of the previously diagnosed mixed mullerian tumor."

4 A reference could be provided for the sentence: "Furthermore, it was reported that DMD deletions, that were detected by SNP array, occurred in 3/7 primary LMS and 8/13 metastatic LMS that abrogated expression of full-length dystrophin transcript, which encodes Dp427m"

5 I think that more attention could be given to how do the LMS subtypes identified in this study correspond with the LMS subtypes described in several previously published transcriptomic studies (Beck et al., Guo et al., Chudasama et al.). Is it possible to identify any universal subtype-specific transcriptomic signatures that could be identified/validated across these independent published datasets? In addition, the UPS-ARL4C associated group in Guo is type II but type I here. Different studies seem to propose their own numbering and selection of markers to define different subtypes. Could there be more consistent and well-defined subtypes identified across multiple studies? I feel that the authors are in a good position to do this and it would be more convincing if there was a thorough comparison with the findings in prior studies.

A reference should be provided for the sentence: "Of note, the established marker for this subtype, ARL4C expression, was overexpressed in Subtype 1 LMS, as expected (Supplementary Figure S8C)."

6 It seems like the RNA-seq data from independent studies were directly integrated in this analysis – how did the authors account for a possible batch effect?

7 The authors do not provide any details on library construction for sequencing. The authors write "Whole-genome and transcriptome (RNA-Seq) sequencing were performed using established protocols on Illumina instruments." – this is not a sufficient level of details. It does not allow e.g. for a comparison of the protocols used for RNA-seq on the Toronto and TCGA cohorts and does not allow to identify, for example, any possible artifacts originating from applying different protocols for gene expression profiling.

8 The references in the text are not numbered in order of appearance

Reviewer #2 (Remarks to the Author): Expert in cancer genomics and evolution

Anderson et al. had studied the origins, development and progression of Leiomyosarcomas (LMS). They have discovered 3 distinct subtypes of LMS that develop from lineages of smooth muscle cells. The authors have performed analysis of WGS and RNA data, including establishing clonal relatedness of lesions and performing timing of specific genomic alterations. Though the paper presents an interesting and novel view into the development of Leiomyosarcomas, some methodological shortcomings would need to be addressed:

1) Clonal analysis and phylogenetic reconstruction needs to be performed at a higher level. A phylogenetic clone base tree reconstruction algorithm needs to be employed (e.g. PhylogicNDT). Do multi-site tumors share subclonal structure? are there changes in signatures between subclones? copy-number events etc?

2) Figure 2a is tough to understand and a better visualization is preferred.

3) Timing of early events can be performed with higher resolution, including ordering most of the important driver events.

4) Real-time (clock) timing needs reevaluation. Specifically with year estimates between neighboring lesions in same patient. Have it been calibrated between lesions with known time of collections? Figure S19 shows quite a broad variation of mutation rate vs age of patient - have these uncertainties been taken into the account? Given number of cases and accuracy of the estimates this section might have to be toned-down appropriately.

5) Direct interpretation of distances vs similarity in UMAP approach is problematic. Can PCA be used instead?

6) What are the novel signatures discovered (S1,S2,S4,S7) refer to? They don't seem to be present or matching SBS/cosmic or PCAWG signatures? Could those be potential artifacts of sequencing? How are they distributed between individual samples? In multiple samples from one patient?

7) Figure S3 and survival analysis. It is unclear if survival differences between subtypes are statically significant. Also the paper states "LMOD1 expression is a strong predictor of overall survival" - has this been tested in a multivariate model with other known predictors or factors that could impact survival? Can a forest plot be added and discussed?

Reviewer #3 (Remarks to the Author): Expert in sarcoma functional genetics

In this study, Anderson et al. studied the molecular landscape of leiomyosarcoma (LMS), which is an aggressive mesenchymal malignancy with limited therapeutic options. Sequencing data from 34 patients were newly generated (44 tumors, 53 whole-genomes, 51 transcriptomes), and additional sequencing data from 79 patients were obtained from TCGA (79 tumors, 17 whole-genomes, 79 transcriptomes). RNA-sequencing (RNA-seq) analysis allocated the LMS cases into three molecular subtypes that differed in patient survival, tumor localization, number of somatic mutations, and transcriptional similarity to the three normal tissue types vascular, digestive, and gynecological smooth muscle, respectively. Furthermore, subtype 1 was transcriptionally associated with myogenic dedifferentiation and M2 macrophage infiltration. A focused search for alterations in eight smooth muscle marker genes in the whole-genome sequencing (WGS) data revealed known recurrent MYOCD amplifications and intragenic deletions in the dystrophin gene (DMD). The latter was associated with lowered DMD full-length transcript (Dp427m) expression and predominant occurrence in subtype 1. Low expression of Dp427m was also found in LMS subtype 3/gynecological LMS in the absence of DMD deletion. Mutational signature analysis of WGS data

showed enrichment of defective homologous recombination repair (HRR) signatures, implying "BRCAness" and sensitivity to drugs targeting this deficiency, which they could validate experimentally. Finally, the authors investigated WGS and RNA-seq data from sequential tumor samples of five patients. These results suggest an early evolutionary divergence of primary and metastatic lesions, and metastatic branching from the primary tumor many years prior to diagnosis.

There are three unique features that differentiate this study from previously reported "LMS genomic landscape" studies: the use of WGS instead of whole-exome sequencing (WES) data, the analysis of sequential tumor samples (diagnosis, metastasis, relapse), and the investigation of multiple biopsies of the same tumor. The latter two analyses suggest the novel finding that LMS tumors develop and diverge decades before diagnosis and progress in parallel. Furthermore, this study provides important confirmatory evidence of previously reported genomic features that were mainly based on WES. Particularly, the report of "BRCAness" mutational signatures based on WES1 needed to be confirmed with more suited WGS data, which has immediate therapeutic impact. However, there are some major issues that need to be clarified and improved.

Major Comments

1. The authors of this study acquired a substantial number of WGS data from LMS patients. It is a pity that this data has not been exploited for an unbiased analysis of the entire LMS genome (maybe even including non-coding regions). The manuscript would substantially benefit from a systematic investigation of all recurrent single nucleotide variants, structural variants, copy number gains and losses, and recording their frequencies. The authors mentioned that some samples are characterized by kataegis, whole-genome duplication, and chromothripsis, which should be reported systematically including frequencies. The latter two phenomena have been shown in LMS before by using WES^{1,2}, but confirmation with WGS, which is the more accurate method for these analyses, would be important.

2. In general, the way how the Figures are cited in the text should be revised. They are often cited at the beginning of the paragraph with multiple panels at once, and when the results are then explained in the text, the corresponding Figure/Figure panel is not cited any more. So the reader is constantly searching for the results in the Figures, which makes the entire manuscript difficult to read and comprehend. In addition, Figure 1E and Suppl. Figure S7 are not cited or somehow mentioned in the text.

3. Discussion: this section needs to be revised substantially. The authors do not discuss their findings in the context of the literature (there is in fact only one paper cited in the entire discussion section). In addition, the authors should not overinterpret their results (e.g. page 18, line 22: "... are the primary determinants ..."; page 20, line 5: "... including dramatic genome-wide duplications, ..."; line 6: "... these late events do not appear to affect patients clinical behavior ...").

4. Page 8-9: in this paragraph, the authors describe recurrent mutations in MYOCD and DMD. Please mention that MYOCD has been described before. The finding of DMD deletions and low Dp427m expression in subtype 3 is novel. Why not showing some of these result in the main figures and mention them in the abstract? Page 8, line 15: please show Figure to this statement ("LMS transcriptomes do have similar expression ...").

5. Are the three molecular subtypes based on transcriptional profiling the same three subtypes that were identified before?^{1,3} Are there similar transcriptional characteristics (e.g. expression of marker genes such as LMOD1, ARL4C)? In addition, are the TCGA and Toronto samples equally distributed in the PCA and UMAP analyses?

6. Page 10, second paragraph: the description of the analysis of the immune microenvironment is very superficial (also in the methods section). What exactly is meant by "... we obtained leukocyte proportion and cell type... "? Are these cell type specific transcriptional profiles or information from histopathology? The higher infiltration of LMS subtype 1 by M2 macrophages should be validated by immunohistochemistry.

7. Page 13: What is the "BRCAness" status of the used primary and established LMS cell lines? The sensitivity to PARP inhibition in the exact same four ATCC cell lines has been tested before.¹ Are the results comparable? In addition, the treated UPS cell lines are not mentioned in the methods section. Please provide some information to these cells. What is CRL 7280 shown in Suppl. Figure S13?

8. Page 14, second paragraph: please provide the data/Figures to these results (line 17: "maintained similar gene expression"; line 19: "... maintained their transcriptional subtype.")

9. Page 15: line 10-11: what is the meaning of the percentages? In general, the authors describe here genetic/genomic alterations that have not been systematically described (see also point 1), which makes the interpretation of this section quite difficult.

Minor Comments:

10. Abstract: please indicate how many samples were newly analyzed and how many were from TCGA.

11. Abstract ("... three specific subtypes develop from distinct lineages of smooth muscle cells."): Since the transcriptional analysis of LMS tumors with the normal tissue types does not prove that the tumors originate from these normal tissues, I want to suggest a more careful wording. This applies also to the Discussion.

12. Abstract lines 12-15 ("... suggesting a novel therapeutic strategy for LMS."): As HRR defects in LMS and sensitivity to PARP inhibition is not a novel finding, this should be rephrased accordingly.

13. Reference #1: the WHO classification of "soft tissue and bone tumors" has been updated this year; this new reference might be used instead of the WHO classification of 2013.

14. The introduction should contain a description of the most prevalent genomic alterations in LMS of previous WES studies, such as recurrent TP53, RB1, and ATRX mutations, MYOCD amplifications, chromothripsis, whole-genome duplication.

15. The naming of the molecular subtypes is inconsistent (e.g. Subtype 1, subtype I, C1, S1), but should always be the same.

16. Page 7, lines 11-12: here, the authors write that subtype 2a consists of abdominal and extremity tumors, and subtype 2b exclusively of abdominal tumors, which is the other way around in Figure 1C. Please clarify.

17. Page 10, line 4-8: the authors describe the usage of transcriptomes from UPS tumors, but the source of these samples is unclear (also not mentioned in the methods section). In addition, please provide a Figure of the clustering of the UPS and LMS subtype 1 tumors.

18. Page 12, line 5: "... including four SBS, ..."; there are 14 SBS signatures in the corresponding Figure 2A.

19. Page 12, line 13-14: "... SBS8, linked to a deficiency in base excision repair."; the SBS8 signature is labeled as unknown in Figure 1A.

20. Page 14, line 7: "... in response to DNA damage ...". How was DNA damage induced?

21. Figure legends need to be revised substantially. Abbreviations are often not explained; important experimental/analyses details are sometimes missing.

22. Figure 1: (B) What are the horizontal black lines? Please add the ID's of the LMS cases to the x-axis. (C) left graph: the choice of the color background makes it difficult to recognize the color of the dots. "UPS-like" is not mentioned anywhere. (D) Please add axis labels. According to the

legend, there should be 1735 dots for the normal tissue, which more looks like some hundred dots.

23. Figure 2: (A) Please add the ID's of the LMS cases to the x-axis. (B) "mM" seems to be a typo. (E) Indicate number of replicates and what is shown (median? Mean? SEM?)

References:

1. Chudasama, P. et al. Integrative genomic and transcriptomic analysis of leiomyosarcoma. *Nature Communications* 9, 1–15 (2018).
2. Network, T. C. G. A. R. et al. Comprehensive and Integrated Genomic Characterization of Adult Soft Tissue Sarcomas. *Cell* 171, 950-953.e28 (2017).
3. Guo, X. et al. Clinically Relevant Molecular Subtypes in Leiomyosarcoma. *Clin Cancer Res* 21, 3501–3511 (2015).

We thank the three reviewers for their valuable questions. The manuscript has benefited tremendously from the peer review and we are eager to share it with the community. Each question or comment has been addressed below in blue text and specific actions taken (typically a new figure, table or analysis) are highlighted in yellow.

Rebecca Gladdy and Adam Shlien

Reviewer #1 (Remarks to the Author): Expert in leiomyosarcoma genomics

This is a very interesting manuscript on an in-depth analysis of leiomyosarcoma, one of the more common forms of sarcoma. The study represents the largest cohort of cases examined by gene expression profiling and whole genome sequencing. The work confirms prior studies by others that indicated that at least three distinct molecular subtypes exist in leiomyosarcoma and provides exciting new data that improves our understanding of this neoplasm.

I am quite enthusiastic about the study but have several questions:

We thank the reviewer for their enthusiasm and constructive feedback, which has helped improve our previously submitted manuscript. Please find below our point-by-point replies to the specific questions provided by this reviewer.

- 1. Page 5, lines 21 – 23: “Full RNA sequencing analysis for expression and fusion gene analysis was also carried out for 130 (51 Toronto, 79 TCGA) transcriptomes using an informatics framework with 97.95% sensitivity and 82.78% accuracy (Fulgini F, in preparation).” – what is actually meant with “informatics framework with 97.95% sensitivity and 82.78% accuracy”? More importantly, can the authors rely on and cite an apparently new method/computational tool that is not yet published and thus is not even described in any detail?**

Thank you for highlighting this issue. We modified the main text for clarity (page 6, line 1). The Materials and Methods has also been updated to outline the exact steps of our RNA-Seq analyses, thereby providing sufficient information to reproduce our work (page 23-25). In addition, we have made clear that the bulk of the bioinformatics pipeline used here are well established and added relevant citations (page 24 lines 4-6; p 25 lines 3-5).

- 2. I wonder whether the sentence “Specifically, an LMS tumor in our cohort resected from an extremity site, clustered with both the subtype 1 group and with normal gynecologic tissue, indicating it was in fact a uterine metastasis.” is perhaps a bit of an overstatement. This suggestion would have carried more weight if there were some clinical support that would suggest that the extremity tumor did indeed originate in the uterus.**

We appreciate this comment. To support our claim that LMS molecular subtyping can be used to reveal a tumor's originating tissue at diagnosis, we have added the relevant clinical information to this section of the discussion (page 20, lines 7-8). This patient did indeed have a clinical history of a primary uterine LMS (not sequenced) and a subsequent metastatic relapse to the leg which was resected, banked and analyzed in this study.

- 3. The cell lines used in this study are very poorly described; the reference provided (J Transl Med 14, 67 (2016)) only mentions one of the 5 cell lines that were newly derived in Toronto. Data should be provided to indicate that these cell lines have maintained smooth muscle differentiation. Regarding the MES-SA cell line – why is this cell line included in this study as an LMS cell line? In ATCC it is described as a uterine sarcoma. In the original publication the tissue of origin was described as “poorly differentiated sarcoma, consistent with elements of the previously diagnosed mixed mullerian tumor.”**

We agree with the information provided by this expert reviewer and apologize for our oversight for including MES-SA as a leiomyosarcoma cell line - this cell line has been removed in its entirety from our revised manuscript. Detailed information regarding the newly derived cell lines generated in Toronto is now provided in **New Supplementary Table 8**. Overall, primary sarcoma cell lines were generated using a standard operating procedure¹ that has been modified to enhance our ability to recover leiomyosarcoma cell lines specifically². Furthermore, in addition to providing our expert pathologist's review of the tumors from which our primary LMS cell lines along with corresponding expression profiles generated from their transcriptomic analysis to indicate that these cell lines maintained their smooth muscle differentiation (**Supplementary Figure 25**). Finally, we did not specifically perform transcriptome analysis of the ATCC cell lines SK-UT-1B and SK-UT-1, as these cell lines underwent whole genome analysis by Chudasama et al³.

- 4. A reference could be provided for the sentence: “Furthermore, it was reported that DMD deletions, that were detected by SNP array, occurred in 3/7 primary LMS and 8/13 metastatic LMS that abrogated expression of full-length dystrophin transcript, which encodes Dp427m”.**

A reference has been provided for this sentence (Page 9, lines 5).

- 5. I think that more attention could be given to how do the LMS subtypes identified in this study correspond with the LMS subtypes described in several previously published transcriptomic studies (Beck et al., Guo et al., Chudasama et al.). Is it possible to identify any universal subtype-specific transcriptomic signatures that could be identified/validated across these independent published datasets? In addition, the**

UPS-ARL4C associated group in Guo is type II but type I here. Different studies seem to propose their own numbering and selection of markers to define different subtypes. Could there be more consistent and well-defined subtypes identified across multiple studies? I feel that the authors are in a good position to do this and it would be more convincing if there was a thorough comparison with the findings in prior studies. A reference should be provided for the sentence: “Of note, the established marker for this subtype, ARL4C expression, was overexpressed in Subtype 1 LMS, as expected (Supplementary Figure S8C).”

We wholeheartedly agree that this manuscript provides an excellent opportunity to review the literature on LMS molecular subtypes. We believe that the data shown here - demonstrating differences in tissue of origin and genomic features between subtypes - provide a strong motivation to generate consensus molecular definitions of LMS. To do so will require a community initiative, as has been done for other malignancies (e.g., medulloblastoma), to discuss the best platforms and to implement appropriate normalization techniques such that data from multiple sites can be cross compared. As suggested by this reviewer, we have now included a discussion on the major studies to date to reconcile similarities and highlight unresolved issues in the molecular subtyping of LMS in our revised discussion (page 19, lines 11-17). Overall, the findings from our work align well with Guo et al, yet the naming convention is not consistent (our type 1 is Guo type 2). However, as stated above, we believe that there is additional collaborative work that is necessary prior to publishing a definitive schema. Thus, a clear next step for our team is to consolidate and clarify the findings from other centers to generate a consensus document that would result in a universal LMS molecular classification system. We would argue that to translate these findings, multiple experts (Pathology, Molecular Biology, Genomics, Medical Oncology etc.) should be engaged in harmonizing these scientific efforts, especially since subtypes likely correlate with outcome and may better stratify the development and response to new therapies. For example, our subtype 1 is poorly differentiated and has increased immune infiltrate, which may warrant investigation with immunotherapy specifically directed at macrophages. Our proposed collaborative group would allow for a foundational approach that would also define how patients are stratified into future clinical trials. Clearly, there is a need to define molecular signatures that are robust and reproducible (immunohistochemistry or transcriptomes), to translate our findings and of others in the management of LMS patients. We look forward to contributing to the development of LMS molecular signatures, as this critical step will advance the field for LMS patients.

A reference has been provided for the suggested sentence (page 10, line 21).

6. It seems like the RNA-seq data from independent studies were directly integrated in this analysis – how did the authors account for a possible batch effect?

We performed several transcriptome-based clustering experiments in our study, all of which included the integration of RNA-seq from Toronto specimens and data from The Cancer Genome Atlas (TCGA). Samples from all sources were analyzed with a common gene expression pipeline, from raw fastq data to expression counts, to prevent the addition of *in silico* noise. For our Principle Component Analysis, we used the `removeBatchEffect` function from the `limma` R package on log transformed data prior to running the clustering. For our orthogonal validations using UMAP and DBSCAN, we applied a step of low variance genes filtering before each dimensionality reduction, which helped reduce dataset differences in the most evident cases. Genes with low variance carry no information useful to the subtyping, while still being affected by batch noise. In all experiments, when mixed together, the samples did not cluster by institution of origin but rather by biological subtype, indicating that this approach was sufficient for a proper identification of characteristic cross-source transcriptional profiles. We have added these details to the Materials and Methods (page 23, lines 28-30; page 24, lines 4-6 and 17-18).

7. The authors do not provide any details on library construction for sequencing. The authors write “Whole-genome and transcriptome (RNA-Seq) sequencing were performed using established protocols on Illumina instruments.” – this is not a sufficient level of details. It does not allow e.g. for a comparison of the protocols used for RNA-seq on the Toronto and TCGA cohorts and does not allow to identify, for example, any possible artifacts originating from applying different protocols for gene expression profiling.

Thank you for this feedback. RNA libraries were constructed using the NEB Ultra II Directional mRNA kit. The resultant libraries were checked on a Bioanalyzer (Agilent) and quantified. New Supplementary Table 11 outlines the RNA starting material and RNA-Seq QC metrics for each sample in the Toronto cohort. Libraries were then multiplexed, clustered and sequenced on a HiSeq2500 in high throughput mode. This information has been added to the Material and Methods (page 23, lines 15-19).

8. The references in the text are not numbered in order of appearance.

We have gone through the manuscript to ensure the in-text references are numbered in order of appearance. We noted one reference on page 4, line 2 (now line 3) (Ref 21) was not being updated using our in-text citation manager, this has been fixed. However, if we referenced a source early in the text and then referenced the same source later in

the text, the original number will appear throughout the manuscript (irrespective of numbering order).

Reviewer #2 (Remarks to the Author): Expert in cancer genomics and evolution

Anderson et al. had studied the origins, development and progression of Leiomyosarcomas (LMS). They have discovered 3 distinct subtypes of LMS that develop from lineages of smooth muscle cells. The authors have performed analysis of WGS and RNA data, including establishing clonal relatedness of lesions and performing timing of specific genomic alterations.

Though the paper presents an interesting and novel view into the development of Leiomyosarcomas, some methodological shortcomings would need to be addressed.

Thank you for the positive overall impression and useful feedback of our methodologies. We addressed each question in detail below. As a brief reminder, in this manuscript we show that LMS can be grouped into three major subtypes; two of these subtypes, dedifferentiated LMS and LMS of gynaecological origin, acquire the highest burden of genomic mutation and are associated with worse survival. We also demonstrate the extremely early origins of LMS and widespread genetic diversity within primary tumors and between metastatic relapses that can be dated back to >20 years before diagnosis. In contrast to more common tumor types, in LMS the significance of genetic factors to subtype, origins or cancer evolution is mostly unknown. Therefore, these novel LMS biomarkers, if validated in a multi-center study, have the potential to be quickly adopted by the sarcoma community.

To our original analyses we have now strengthened the text and modified figures in multiple places, especially to highlight the novelty of our findings and improve the clarity of our main messages.

- 1. Clonal analysis and phylogenetic reconstruction needs to be performed at a higher level. A phylogenetic clone base tree reconstruction algorithm needs to be employed (e.g. PhylogicNDT). Do multi-site tumors share subclonal structure? are there changes in signatures between subclones? copy-number events etc?**

Thank you for this important suggestion. We have now performed a higher-resolution clonal analysis of LMS by applying two complimentary tools: MutationTimeR⁴; and DPCLust^{5,6}. MutationTimeR can time somatic mutations relative to clonal and subclonal copy number states and calculate the relative timing of copy number gains, while DPCLust uses a Dirichlet process to provide information on clonal/subclonal cancerous cells defined by distinct somatic mutations.

In doing so, we found that the DPCLust-based subclonal reconstruction of LMS cancer evolution corroborated our original tree constructions while providing finer detail of the populations present in each tumor region and paired primary-relapse specimens. All samples from the same patient had a founder clone that harbored the most recurrent drivers of LMS (*TP53* alterations and *ATR*X deletions). Each tumor region then went on to acquire 1-3 major clones but continued to diversify and acquire unique subclonal mutations. With respect to copy-number events, our major finding is that multiple arm-level or whole chromosome losses are one of the defining features of the LMS founder clonal population (and thus shared). The most frequent genes in these losses involve known LMS drivers, *TP53* (17p), *RB1* (chr 13q) and *PTEN* (chr 10q). There are further copy-number alterations that arise throughout tumor evolution, but aside from these regions of the genome, they are not as recurrent across samples. We have modified Figures 3 and 4 by adding colored circles to represent the major and minor clones within each sample, and to highlight the clonal dynamics of mutations in LMS. Additionally, we have generated new supplementary figures of patients with at least 3 samples to illustrate the changes in cancer cell fraction of distinct clonal populations of LMS and how they fluctuate (page 17 lines 5-8, New Supplementary Figures 21-23).

To evaluate subclonal mutational signatures, we used DPCLust-based reconstruction to identify subclones (CCF < 0.7). Based on data from Figure 2, SigProfiler provides the probabilities that a given mutational process causes each mutation type at a given trinucleotide context. We used these probabilities to assign mutational signatures to each mutation, which was classified as either clonal or subclonal. There was not a significant difference between the relative proportion of mutational signatures (clonal or subclonal) between regions of the same tumor. The greatest difference in mutational signature proportions was observed between primary and metastatic relapses. We have included a representative example of patient Ab6 (below) showing variations in the overall number of mutations assigned to different mutational signatures.

R2 #1 Figure for Reviewers. Subclonal signatures in Ab6. A representative example of signatures across Ab6's samples (primary [Ab6T] and metastatic lesions [Ab6Met1T, Ab6MuRe1-5T]) in subclones (CCF < 0.7).

2. Figure 2a is tough to understand and a better visualization is preferred.

Thank you for highlighting this issue. We have improved figure 2a in multiple ways, including:

- a. Adding sample name labels to the x-axis
- b. Collapsing infrequent SBS signatures into the "Other" category for better readability
- c. Increasing font size of all text
- d. Moving the figure legend up next to Fig 2a
- e. Changing the "Molecular Etiology" colors to blue to easily distinguish signature names from suggested causation

3. Timing of early events can be performed with higher resolution, including ordering most of the important driver events.

We thank the reviewer for this suggestion. As a reminder, the most important driver events of LMS include somatic *TP53*, *RB1* and *ATRX* mutations. Other genomic events that have been documented in LMS include genome duplications and chromothripsis. In our initial submission, we showed that *TP53* and *RB1* mutations were always common

to the trunk of our multi-region tumors and/or primary-relapse pairs. Together with its near-universal disruption, this provides further evidence that these variants are likely obligate events required for LMS formation. Additionally, we report arm-level and whole chromosome LOH events of 10q, 13q and 17p that precede genome doubling, co-occurring in evolutionary time with *TP53* mutations. Lastly, we show that chromothripsis and kataegis events were unique to individual primary or relapse tumors, suggesting these are mid-late evolutionary events. Thus, we have determined the relative ordering of the important driver events in this cancer type. This is now further validated by our new MutationTimeR analysis (page 15 lines 8-9). To highlight these findings, we have provided further detailed annotations to the trees in Figures 3 and 4, as well as Supplementary Figure 16 and **new Supplementary Figure 21-23**. We have also generated a working model of leiomyosarcomagenesis that is featured in updated Supplementary Figure 18D.

- 4. Real-time (clock) timing needs reevaluation. Specifically with year estimates between neighboring lesions in same patient. Have it been calibrated between lesions with known time of collections? Figure S19 shows quite a broad variation of mutation rate vs age of patient - have these uncertainties been taking into the account? Given number of cases and accuracy of the estimates this section might have to be toned-down appropriately.**

We agree with the reviewer and thank them for addressing these points that need revision. We have added Part B to supplementary figure S19 (now **new Supplementary Figure 24**) which illustrates a representative example of the calibration we performed between two lesions collected at different times points from the same patient. We demonstrate how to calculate the SBS1 mutation rate and predict the number of SBS1 mutations at relapse (and compare the calculated value to the actual number of SBS1 mutations in the relapse that were extracted from our mutational signature analysis are close). Given the number of cases (n=5, abdominal only), we agree with the reviewer that the limitations of this analysis should be discussed. We have added these points to the main text (page 20, lines 20-21).

- 5. Direct interpretation of distances vs similarity in UMAP approach is problematic. Can PCA be used instead?**

The reviewer is correct that non-linear dimensionality reduction algorithms like UMAP do not conserve distances during the transformation. Rather, UMAP attempts to conserve neighborhood relations between the projected points. While this can be sufficient for clusters identification with density-based methods, it may still be problematic when attempting to associate new samples to the classes by projecting and

measuring their distances from the reference samples. To limit this issue, we worked in a medium-dimensionality target space (12D), on which the extremely-highly-dimensional source space of the full transcriptome was projected. This approach can limit the deformation of space while still allowing us to work with manageable number of dimensions. Furthermore, the class assignment by weighted k-NN does not rely on single pair distances, but rather averages the assignment across the k-closest points. If we assume the conservation of nearest-neighbors is valid, this method can give sufficiently robust results when points fall in the proximity of one or more classes.

While PCA is a valid alternative to these methods, it has two main limitations. First, it is linear and unable to capture more complex relationships across features. Second, the features removal step can effectively discard a considerable amount of information. Non-linear methods, like t-SNE or UMAP attempt to maintain all information in the input and use it to determine position of each point in the final space.

6. What are the novel signatures discovered (S1,S2,S4,S7) refer to? They don't seem to be present or matching SBS/cosmic or PCAWG signatures? Could those be potential artifacts of sequencing? How are they distributed between individual samples? In multiple samples form one patient?

As a reminder, our mutational signature analysis involved a *de novo* extraction approach using two established signature extraction tools: SigProfiler; and SigAnalyzer. The novel signatures that were discovered by SigAnalyzer are likely not potential artifacts of sequencing as they are present in multiple samples from different patients (see figure below).

Instead, we believe these are real, established mutational signatures in LMS genomes that were not correctly matched to the COSMIC or PCAWG reference signatures. On the other hand, no novel signatures were detected by SigProfiler (that is, all signatures matched to COSMIC/PCAWG).

Since our analysis only used SigProfiler results, and to avoid any further confusion for the reader, we have removed mention of our SigAnalyzer analysis in the main text and have removed figure S9.

R2 #6 Figure for Reviewers. SigAnalyzer LMS Signatures. Mutation signature discovery by BayesNMF, a Bayesian variant of the SigAnalyzer NMF algorithm, for LMS whole-genome sequences.

7. **FiugreS3 and survival analysis. It is unclear if survival differences between subtypes are statically significant. Also the paper states "LMOD1 expression is a strong predictor of overall survival" - has this been tested in a multivariate model with other known predictors or factors that could impact survival? Can a forest plot be added an discussed?**

Overall, in our study, it is apparent that there is a biologic difference between subgroups 1, 2 and 3. Based on the data provided in our cohort we were not able to show that this was statistically significant between the three groups. However, when we grouped subtypes 1 and 3 together, each of which trended towards inferior survival compared to subtype 2, we did see a significant difference. To specifically address this query regarding a multivariate model, we have generated the forest plot with other known factors of survival (see below). Our analysis suggests that both anatomical location and

LMOD1 expression are independent determinants of patient survival – these features are highlighted in our molecular subtypes. We believe that external validation of these subtype findings is critical to the development of a molecular classification system for LMS (please see response to Reviewer 1, point #5). Thus, a future direction of our work is to collaborate with other LMS researchers to ensure that an adequately powered study is performed, which is always a challenge in rare tumors.

R2 #7 Figure for Reviewers. Prognostic factors in LMS. Forest plot of factors associated with poor survival in LMS. Anatomical location and *LMOD1* expression may contribute LMS survival.

Reviewer #3 (Remarks to the Author): Expert in sarcoma functional genetics

In this study, Anderson et al. studied the molecular landscape of leiomyosarcoma (LMS), which is an aggressive mesenchymal malignancy with limited therapeutic options. Sequencing data from 34 patients were newly generated (44 tumors, 53 whole-genomes, 51 transcriptomes), and additional sequencing data from 79 patients were obtained from TCGA (79 tumors, 17 whole-genomes, 79 transcriptomes). RNA-sequencing (RNA-seq) analysis allocated the LMS cases into three molecular subtypes that differed in patient survival, tumor localization, number of somatic mutations, and transcriptional similarity to the three normal tissue types vascular, digestive, and gynecological smooth muscle, respectively. Furthermore, subtype 1 was transcriptionally associated with myogenic dedifferentiation and M2 macrophage infiltration. A focused search for alterations in eight smooth muscle marker genes in the whole-genome sequencing (WGS) data revealed known recurrent MYOCD amplifications and intragenic deletions in the dystrophin gene (DMD). The latter was associated with lowered DMD full-length transcript (Dp427m) expression and predominant occurrence in subtype 1. Low expression of Dp427m was also found in LMS subtype 3/gynecological LMS in the absence of DMD deletion. Mutational signature analysis of WGS data showed enrichment of defective homologous recombination repair (HRR) signatures, implying “BRCAness” and sensitivity to drugs targeting this deficiency, which they could validate experimentally. Finally, the authors investigated WGS and RNA-seq data from sequential tumor samples of five patients. These results suggest an early evolutionary divergence of primary and metastatic lesions, and metastatic branching from the primary tumor many years prior to diagnosis.

There are three unique features that differentiate this study from previously reported “LMS genomic landscape” studies: the use of WGS instead of whole-exome sequencing (WES) data, the analysis of sequential tumor samples (diagnosis, metastasis, relapse), and the investigation of multiple biopsies of the same tumor. The latter two analyses suggest the novel finding that LMS tumors develop and diverge decades before diagnosis and progress in parallel. Furthermore, this study provides important confirmatory evidence of previously reported genomic features that were mainly based on WES. Particularly, the report of “BRCAness” mutational signatures based on WES1 needed to be confirmed with more suited WGS data, which has immediate therapeutic impact. However, there are some major issues that need to be clarified and improved.

We appreciate the reviewer’s constructive comments, which we address in detail below. We thank the reviewer for recognizing our unique cohort of LMS patients and the work that went into developing and optimizing the informatics analysis. We have made many changes to the manuscript to better highlight the impact of our results.

Major Comments

- 1. The authors of this study acquired a substantial number of WGS data from LMS patients. It is a pity that this data has not been exploited for an unbiased analysis of the entire LMS genome (maybe even including non-coding regions). The manuscript would substantially benefit from a systematic investigation of all recurrent single nucleotide variants, structural variants, copy number gains and losses, and recording their frequencies. The authors mentioned that some samples are characterized by kataegis, whole-genome duplication, and chromothripsis, which should be reported systematically including frequencies. The latter two phenomena have been shown in LMS before by using WES^{1,2}, but confirmation with WGS, which is the more accurate method for these analyses, would be important.**

We agree that there is a great wealth of information provided by whole genome sequence data. While the driver landscape of LMS has been reported by prior exome studies^{3,7}, in this manuscript we define this tumor's subtypes, origins and somatic evolution. We did so by using the full gamut of variants present in cancer genomes – both coding and non-coding. For instance, 99% of the substitutions used for mutational signature are from non-coding regions.

As suggested by the Reviewer, we have now included new supplemental tables detailing the recurrent events, including small substitutions and indels (new Supplementary Table 1-2), rearrangements (new Supplementary Table 3), large-scale copy number changes (new Supplementary Table 4), and regions of kataegis and chromothripsis (new Supplementary Tables 5-6). To enable a future explorations of the driver landscape of LMS all of our genome data has been made available through the European Genome-phenome Archive (EGA).

- 2. In general, the way how the Figures are cited in the text should be revised. They are often cited at the beginning of the paragraph with multiple panels at once, and when the results are then explained in the text, the corresponding Figure/Figure panel is not cited any more. So the reader is constantly searching for the results in the Figures, which makes the entire manuscript difficult to read and comprehend. In addition, Figure 1E and Suppl. Figure S7 are not cited or somehow mentioned in the text.**

We thank the reviewer for this suggestion. Where appropriate, we have modified how the figures are cited in the manuscript, so the reader can more easily comprehend the main messages of the paper. We would be happy to work with the Editor to ensure the figures are referenced and labeled correctly.

Figure 1E and Figure S7 were referenced in the original main text on page 10, line 3.

“Consistent with this notion, most markers of muscle differentiation, including *leimodin1* (*LMOD1*), *caldesmon* (*CALD1*), and *myocardin* (*MYOCD*) were diminished in subtype 1 (Fig. 1E, Supplementary Fig. 7A).”

3. **Discussion: this section needs to be revised substantially. The authors do not discuss their findings in the context of the literature (there is in fact only one paper cited in the entire discussion section). In addition, the authors should not overinterpret their results (e.g. page 18, line 22: “... are the primary determinants ...”; page 20, line 5: “... including dramatic genome-wide duplications, ...”; line 6: “... these late events do not appear to affect patients clinical behavior ...”).**

We thank the reviewer for highlighting this. We had ensured that prior manuscripts, which have been foundational to our own, were cited as needed in the introduction and throughout the results. We have modified the discussion to discuss our novel results in the context of literature. This includes a statement on shared LMS molecular subtypes and references to prior functional work. The wording has been toned down where suggested.

4. **Page 8-9: in this paragraph, the authors describe recurrent mutations in *MYOCD* and *DMD*. Please mention that *MYOCD* has been described before. The finding of *DMD* deletions and low *Dp427m* expression in subtype 3 is novel. Why not showing some of these result in the main figures and mention them in the abstract? Page 8, line 15: please show Figure to this statement (“*LMS* transcriptomes do have similar expression ...”).**

As suggested by the reviewer, we have included a description of the most prevalent genomic alterations in LMS, including *MYOCD* amplifications, in the introduction (page. 3, Line 18-20).

We agree that the finding of *DMD* deletions and low *Dp427m* expression in subtype 3 is an interesting result that should be highlighted appropriately. We have added this information to the abstract (page 2, line 11). Additionally, we have generated a new supplementary figure to show that LMS tumors with *DMD* deletions have similar *DMD* expression to muscular dystrophy patients (new Supplementary Figure 7). In revising this manuscript, we have decided to provide this data as supplemental primarily due to space constraints but appreciate the interest and agree understanding the regulation of these key myogenic genes in LMS warrants further investigation.

5. **Are the three molecular subtypes based on transcriptional profiling the same three subtypes that were identified before?1,3 Are there similar transcriptional**

characteristics (e.g. expression of marker genes such as LMOD1, ARL4C)? In addition, are the TCGA and Toronto samples equally distributed in the PCA and UMAP analyses?

Overall, the three subtypes have been mostly recapitulated in our transcriptomic analysis with reference 3 but not fully with reference 1. As per our rebuttal to Reviewer 1, point 5, there are several reasons for discrepancies which likely arise from two issues: previous methodology used (whole transcriptome vs not); and the spectrum of clinical specimens evaluated along with the extent of clinical data available. The three molecular subtypes have a pattern related to anatomic site, yet this is not resolute. Thus, we would favor that a working group be established to resolve these seminal issues such that nomenclature, clinical information and biomarkers are developed that will be clinically relevant and useful for patient stratification and possibly prognostication.

- 6. Page 10, second paragraph: the description of the analysis of the immune microenvironment is very superficial (also in the methods section). What exactly is meant by "... we obtained leukocyte proportion and cell type... "? Are these cell type specific transcriptional profiles or information from histopathology? The higher infiltration of LMS subtype 1 by M2 macrophages should be validated by immunohistochemistry.**

For our immune microenvironment investigation, we performed an *in silico* analysis based on the supplementary data provided in NIHMS958212-supplement-2 from Thorsson et al.⁸. We have added this information to the main text and Materials and Methods (page 10, line 16 and page 26, line 2). The data from this publication is derived from DNA methylation probes, while the relative fraction of the 22 immune cell types within the leukocyte compartment were estimated using CIBERSORT.

To further support our data analysis, a recent paper by Dancsok et al. (senior author E. Demicco, corresponding author from TCGA)⁹, characterized the extent of macrophage subtypes in 1242 sarcomas, from over 27 subtypes. In TMAs from 20 LMS specimens there was a higher proportion of macrophages compared to tumor infiltrating lymphocytes (TILs) and the median CD68 cell count was 273/mm² and CD163 was 281/mm². They also found independently similar proportions of macrophage to immune infiltrate for LMS with a striking range (median 0.5, range 0.2-0.9; their Figure 2e), by mRNA expression signatures calculated on TCGA sarcoma subtypes. Unfortunately, there are no TMAs available for further immunohistochemical validation from TCGA sarcoma (personal communication, Dr. E. Demicco.)

Finally, in response to this review, we have reviewed the H&E stains to confirm diagnosis / lesional tissue and avoided quantifying in areas of necrosis and inflammatory aggregates. Next, we evaluated commonly used macrophage markers CD68 as a marker

of M1-like (pro-inflammatory) macrophages and CD163 as a marker of M2-like (anti-inflammatory) macrophages, which were scored blinded to subtype by an expert sarcoma pathologist. Immunohistochemistry with the macrophage polarized markers (Clones: CD68 PG-M1; CD163 MRQ-26) was performed on a subset of our LMS specimens representative of each molecular subtype (subtype 1 n=6, subtype 2 n=3 and subtype 3, n=5). IHC stain quality was confirmed by the evaluation of on-slide positive controls. An effort was made to only quantify cells with a macrophage morphology (not tumour cells expressing the marker). Regrettably, we are not able to provide this data for our entire cohort due to limited specimens that are available from our biobank. Overall, we saw heterogeneity within each specimen, a common feature found in the Dancsok paper. In our analysis, which is supportive of our computational findings, we see that overall, there is a higher number of M2-like macrophages in subtype 1, especially in the gynecologic specimens (see Figure 1 for Reviewers only) yet with a small data set, any formal conclusions are premature. Thus, we would agree that along with our previous statements (see reviewer 1, point 3, reviewer 2, point 7 and this reviewer point 3) comprehensive additional studies that validate the immune landscape of each subtype is critical, and using higher resolution assays such as tissue CyTOF are likely required. In these future studies, immunoprofiling these heterogenous tumors in addition to defining molecular markers and prognostic variables will be critical to design immunotherapy clinical trials beyond checkpoint blockade, which has had limited success in LMS to date.

R3 #6 Figure for Reviewers. Quantification of tumor-associated macrophages in LMS.

(A) Boxplots depicting comparative counts of CD68+ macrophages (M1), CD163+ macrophages (M2) across LMS subtypes. Abbreviations; A/E, abdominal/extremity; Gyn, gynecological. (B) Immunohistochemical staining of CD68 (M1-like macrophage) and CD163 (M2-like macrophage) (Clones: CD68 PG-M1; CD163 MRQ-26) in our three LMS subtypes.

7. Page 13: What is the “BRCAness” status of the used primary and established LMS cell lines? The sensitivity to PARP inhibition in the exact same four ATCC cell lines has been tested before.¹ Are the results comparable? In addition, the treated UPS cell lines are not mentioned in the methods section. Please provide some information to these cells. What is CRL 7280 shown in Suppl. Figure S13?

- a. Unfortunately, the *BRCAness* status is unknown in cell lines as variant calling (and thus mutational signature detection) is not reliable without a matched normal, which is not available for some of our cell lines and certainly not for ATCC cell lines. As a proof-of-principle, we identified primary tumor samples with high levels of SBS3 and/or ID6 (substitution and indel HRD signatures) and purposefully mismatched (MM) the corresponding normal during the variant calling step. We then filtered against a panel of unrelated normals and ran our mutational signature pipeline to determine if SBS3 or ID6 were detected. HR-deficiency signatures were not detectable (see figure below).

R3 #7a Figure for Reviewers. HR-Signatures in Samples with no matched normal.

Substitution (A) and indel (B) signature activity in 3 HR-deficient LMS samples with a matched normal (Ut2T,

Ab11Met3RTT, and Ab10RTT) and with a purposefully mismatched normal (Ut2T MM, Ab11Met3RTT MM, Ab10RTT MM). (A) Heatmap shows that with the mismatched samples present, SBS3 is not detected. (B) Barplot shows that ID6 is present (pink bars, red arrow) with the correct matched arrow, but disappears in the purposefully mismatched normal samples.

- b. With respect to the PARP inhibition assays, in our study, we used a stricter definition of sensitivity (<1uM) for olaparib. Thus, in our Fig 2b., most LMS cell lines were scored as either partially responsive or non-responsive to this particular compound, which is different from previously published studies that had a higher sensitivity threshold (most often~ 5uM) when used as a single agent. This is now highlighted in the discussion for reader clarity (page 21, line 19-20). Previously four uterine ATCC cell lines were tested (including MES-SA, no longer in this manuscript), however in our current study we used an expanded panel of primary LMS cell lines (n=5), derived from additional LMS anatomical sites (please see Reviewer 1, point 3). Furthermore, compound addition was performed digitally using 24-point serial dilutions (0.013-10µM) and viability was assayed with ATP-lite, which has enhanced sensitivity compared to clonogenic assays. Finally, we tested other DDRi of which we demonstrated sensitivity of the majority of LMS cells to CHK1 and WEE1 inhibitors.
 - c. We thank the reviewer for detecting that the treated UPS cell lines were not mentioned in the methods section and we have rectified this and provided more information on these specimens in the text (page 26, line 29, **new Supplementary Table 9**)
 - d. The cell line used in Supplementary Figure S13 is CRL-7518. We have added more information on this cell line in the Materials and Methods, as well as the figure legend.
- 8. Page 14, second paragraph: please provide the data/Figures to these results (line 17: “maintained similar gene expression”; line 19: “... maintained their transcriptional subtype.”)**
We have modified the UMAP in supplementary figure 4 to highlight paired primary-metastatic relapse samples that cluster together and maintain their transcriptional subtype.
- 9. Page 15: line 10-11: what is the meaning of the percentages? In general, the authors describe here genetic/genomic alterations that have not been systematically described (see also point 1), which makes the interpretation of this section quite difficult.**

The percentages in this paragraph of the manuscript refer to the frequency of deletion events in the tumor suppressor genes listed that precede whole-genome duplication.

For example, in samples that harbor WGD events, 93% of LOH of *RB1* precedes WGD. For clarity, we've added the frequency of WGD cases from which the numbers are derived (page 15, lines 7-8). Further, we have included **new Supplementary Table 4** which includes allele-specific copy number frequencies for the genes in question.

Minor Comments:

10. Abstract: please indicate how many samples were newly analyzed and how many were from TCGA.

We appreciate this request but are concerned about the complexity of adding these granular details in the abstract itself where we attempt to highlight the major findings of the paper. We would direct the reader Figure S1 which comprehensively annotates the combined study cohort.

11. Abstract (“ ... three specific subtypes develop from distinct lineages of smooth muscle cells.”): Since the transcriptional analysis of LMS tumors with the normal tissue types does not proof that the tumors originate from these normal tissues, I want to suggest a more careful wording. This applies also to the Discussion.

We would agree that our informatics approach does not directly prove that distinct LMS subtypes arise from normal tissue types. However, with the ability to compare bulk tumor whole transcriptomes with comprehensive expression levels of human normal tissue, this allows one to more closely correlate a tumor from a specific anatomic site with cell lineage.

We have more carefully rephrased as suggested in the abstract to: “We uncovered three specific subtypes of LMS that likely develop from distinct lineages of smooth muscle cells” (page 2, lines 8-9). This is a reasonable developmental biology hypothesis based on the current state of whole transcriptome data, which our group will continue to explore and refine.

12. Abstract lines 12-15 (“... suggesting a novel therapeutic strategy for LMS.”): As HRR defects in LMS and sensitivity to PARP inhibition is not a novel finding, this should be rephrased accordingly.

We thank the reviewer for ensuring that we highlight the novelty of our functional studies. In fact, as per this reviewer major point 7, in Fig. 2b, with the definitions we used for sensitivity and assays performed across cell lines from different LMS sites, we found the majority of our cell lines when used as a single agent, LMS cell lines were either olaparib partially or not responsive. We have modified the abstract to include the word PARP ‘trappers’ (page 2, line 14). In our study, we also have included other DDRi

(ATR, WEE1 and CHEK1), the former of which is currently under investigation in other soft tissue sarcomas, in combination, i.e. Phase 2 expansion AZD1775 with irinotecan for rhabdomyosarcoma¹⁰.

13. Reference #1: the WHO classification of “soft tissue and bone tumors” has been updated this year; this new reference might be used instead of the WHO classification of 2013.

This reference is updated in accordance with the 5th edition of the WHO soft tissue and bone tumors, which was released earlier this year (page 3, line 3).

14. The introduction should contain a description of the most prevalent genomic alterations in LMS of previous WES studies, such as recurrent TP53, RB1, and ATRX mutations, MYOCD amplifications, chromothripsis, whole-genome duplication.

A description of the most prevalent genomic alterations in LMS has been added to the introduction (page 3, lines 18-20).

15. The naming of the molecular subtypes is inconsistent (e.g. Subtype 1, subtype I, C1, S1), but should always be the same.

We agree with the reviewer that molecular subtype naming conventions should be consistent throughout the paper. We have modified all supplementary figures and legends that reference the subtypes irregularly.

16. Page 7, lines 11-12: here, the authors write that subtype 2a consists of abdominal and extremity tumors, and subtype 2b exclusively of abdominal tumors, which is the other way around in Figure 1C. Please clarify.

We thank the reviewer for noticing this error. This has been revised in the main text to reflect the Figure 1C, which was correct (page7, lines 11-12, 23, page 8 line 1).

17. Page 10, line 4-8: the authors describe the usage of transcriptomes from UPS tumors, but the source of these samples is unclear (also not mentioned in the methods section). In addition, please provide a Figure of the clustering of the UPS and LMS subtype 1 tumors.

We agree with the reviewer that the source of UPS specimens is important information. The UPS tumors used in the clustering analysis were acquired from the UCSC Treehouse Childhood Cancer Initiative, which includes datasets from TCGA and St. Jude’s Children’s Hospital. We have modified the text to make the source of this data clearer (page 10, line 7). As suggested by the reviewer, we have generated a new supplementary figure to

show UPS tumors clustering with LMS subtype 1 cancers (new Supplementary Figure 9).

- 18. Page 12, line 5: “... including four SBS, ...”; there are 14 SBS signatures in the corresponding Figure 2A.**

We thank the reviewer for noticing this error. This has been revised in the main text (page 12, line 5).

- 19. Page 12, line 13-14: “... SBS8, linked to a deficiency in base excision repair.”; the SBS8 signature is labeled as unknown in Figure 1A.**

Unlike the other mutational signatures in Figure 2A, the mechanistic basis of SBS8 is still debated. The study that we cite here¹¹, reports that SBS8 is elevated in nucleotide-excision repair deficient mice, organoid cultures and NER-deficient breast cancers. However, this has yet to be externally validated. Therefore, the definitive molecular etiology of SBS8 is unknown, although it has been putatively linked to this mutagenic source.

- 20. Page 14, line 7: “... in response to DNA damage ...”. How was DNA damage induced?**

This genome engineering system uses rare-cutting endonucleases to exploit endogenous DNA repair pathways¹². We have added this information to the main text (page 14, line 4-5).

- 21. Figure legends need to be revised substantially. Abbreviations are often not explained; important experimental/analyses details are sometimes missing.**

Thank you for highlighting this issue. We have addressed this comment by spelling out acronyms and double-checking every figure legend to ensure sufficient experimental/analytical details have been outlined. We are happy to work with editor to refine this further.

- 22. Figure 1: (B) What are the horizontal black lines? Please add the ID’s of the LMS cases to the x-axis. (C) left graph: the choice of the color background makes it difficult to recognize the color of the dots. “UPS-like” is not mentioned anywhere. (D) Please add axis labels. According to the legend, there should be 1735 dots for the normal tissue, which more looks like some hundred dots.**

- a. Figure 1B: The horizontal black lines refer to the median value of the subtype. This information has been added to the figure legend.

- b. Figure 1B: We appreciate the request to add sample IDs of the LMS cases to the x-axis, however we hesitate to add this level of detail to this figure as the minimum font size required for legibility would not be achieved. We therefore direct the reviewer to **new Supplementary Table 7**, which provides the samples IDs in Figure 1 and the corresponding genomic mutation burdens.
- c. Figure 1C: The color choice of the background reflects the color schemes of the subtypes.
- d. Figure 1C: We have removed the term “UPS-like” from the figure and replaced it with “dedifferentiated”
- e. Figure 1D: We’ve added the UMAP-1 and UMAP-2 axis labels back to the plot, however there is no interpretability to the axes of a UMAP.
- f. Figure 1D: We thank the reviewer for highlighting this error. The analysis included 1735 normal tissues from GTEx, however only the smooth-muscle tissues that cluster with LMS are plotted here (n=271). We have corrected the figure legend.

23. Figure 2: (A) Please add the ID’s of the LMS cases to the x-axis. (B) “mM” seems to be a typo. (E) Indicate number of replicates and what is shown (median? Mean? SEM?).

- a. Figure 2A: In accordance with Reviewer 2 Point 2, we have modified Figure 2A in several ways, including adding sample names to the x-axis.
- b. Figure 2B: We thank the reviewer for detecting this error. We have corrected this in the figure.
- c. Figure 2E: The bar plot illustrates the results from 3 independent experiments. The percentage of measured events refer to the proportion of HR events (GFP signal) to NHEJ (mCherry signal) in each LMS cell line and controls, with the standard error of the mean (SEM).

References

1. Chudasama, P. et al. Integrative genomic and transcriptomic analysis of leiomyosarcoma. *Nature Communications* 9, 1–15 (2018).
2. Network, T. C. G. A. R. et al. Comprehensive and Integrated Genomic Characterization of Adult Soft Tissue Sarcomas. *Cell* 171, 950-953.e28 (2017).
3. Guo, X. et al. Clinically Relevant Molecular Subtypes in Leiomyosarcoma. *Clin Cancer Res* 21, 3501–3511 (2015).

Point-By-Point References

1. Okada, T. *et al.* Integrin- α 10 dependency identifies RAC and rictor as therapeutic targets in high-grade myxofibrosarcoma. *Cancer Discov.* (2016) doi:10.1158/2159-8290.CD-15-1481.
2. Tanaka, T., Toujima, S., Toyoda, S., Takeuchi, S. & Umesaki, N. Establishment and characterization of novel human uterine leiomyosarcoma cell lines. *Int. J. Oncol.* (2010) doi:10.3892/ijco-00000660.
3. Chudasama, P. *et al.* Integrative genomic and transcriptomic analysis of leiomyosarcoma. *Nat. Commun.* (2018) doi:10.1038/s41467-017-02602-0.
4. Gerstung, M. *et al.* The evolutionary history of 2,658 cancers. *Nature* (2020) doi:10.1038/s41586-019-1907-7.
5. Nik-Zainal, S. *et al.* The life history of 21 breast cancers. *Cell* (2012) doi:10.1016/j.cell.2012.04.023.
6. Dentre, S. C., Wedge, D. C. & Van Loo, P. Principles of Reconstructing the Subclonal Architecture of Cancers. *Cold Spring Harbor perspectives in medicine* (2017) doi:10.1101/cshperspect.a026625.
7. Abeshouse, A. *et al.* Comprehensive and Integrated Genomic Characterization of Adult Soft Tissue Sarcomas. *Cell* (2017) doi:10.1016/j.cell.2017.10.014.
8. Thorsson, V. *et al.* The Immune Landscape of Cancer. *Immunity* (2018) doi:10.1016/j.immuni.2018.03.023.
9. Dancsok, A. R. *et al.* Tumor-associated macrophages and macrophage-related immune checkpoint expression in sarcomas. *Oncoimmunology* (2020) doi:10.1080/2162402X.2020.1747340.
10. Cole, K. A. *et al.* Phase I clinical trial of the WEE1 inhibitor adavosertib (AZD1775) with irinotecan in children with relapsed solid tumors: A COG phase I consortium report (ADVL1312). *Clin. Cancer Res.* (2020) doi:10.1158/1078-0432.CCR-19-3470.
11. Jager, M. *et al.* Deficiency of nucleotide excision repair is associated with mutational signature observed in cancer. *Genome Res.* (2019) doi:10.1101/gr.246223.118.
12. Certo, M. T. *et al.* Tracking genome engineering outcome at individual DNA breakpoints. *Nat. Methods* (2011) doi:10.1038/nmeth.1648.

REVIEWER COMMENTS

Reviewer #1 (Remarks to the Author):

The authors have completely addressed all my concerns in this revised manuscript.

Reviewer #2 (Remarks to the Author):

I commend the authors for addressing most of previous concerns.
A few points remain that could be improved:

1) There seem to be some potential issues with copy number reconstruction based on the clonal plots in the paper. It would be good if authors can provide those (as Supplement) as an additional quality control.

2) There is some concern with the tree reconstruction results from Treeomics. Is there an alternative methods authors can use to validate the results? Additionally, it would be really useful to depict branch length and mutation numbers assigned to each branch of the tree and some measure of sample composition. Are all samples arise from independent branches (i.e. single cell seeding) ? For multiregional tumors that seems to be an unlikely scenario. It is more likely that samples with some proximity to each other share subclones.

Reviewer #3 (Remarks to the Author):

The authors have addressed all of my comments and concerns.

One minor comment: It would be helpful to replace "CRL-7518" (which is the ATCC order number) on page 13 with the actual name of the cell line (Hs 789.Sk) and mention that this is a normal skin cell line. Accordingly, the M&M section should be adapted.

Reviewer #1 (Remarks to the Author):

The authors have completely addressed all my concerns in this revised manuscript.

Reviewer #2 (Remarks to the Author):

I commend the authors for addressing most of previous concerns.

A few points remain that could be improved:

- 1. There seem to be some potential issues with copy number reconstruction based on the clonal plots in the paper. It would be good if authors can provide those (as Supplement) as an additional quality control.**

As suggested, the copy number plots are included in a new supplemental figure. The full copy number reconstruction plots for all seven sequenced genomes for this patient (Ab11) are shown (new supplementary figure 26).

- 2. There is some concern with the tree reconstruction results from Treeomics. Is there an alternative methods authors can use to validate the results? Additionally, it would be really useful to depict branch length and mutation numbers assigned to each branch of the tree and some measure of sample composition. Are all samples arise from independent branches (i.e. single cell seeding)? For multiregional tumors that seems to be an unlikely scenario. It is more likely that samples with some proximity to each other share subclones.**

We thank the reviewer for their valuable feedback which has greatly improved the quality of our manuscript. We recognize the shortcomings of Treeomics since it does not perform clonal resolution of phylogenies. As suggested, we addressed this by using a clone-based reconstruction tool that leverages the multi-site samples and primary-relapse pairs we collected. This tool, DPClust, uses a Dirichlet process to reconstruct tumour subpopulations based on their cancer cell fractions (CCF). We are pleased that this analysis independently recapitulated the trees generated by Treeomics, thus validating our results. It has also considerably increased the resolution with which we could map the mutation evolution of LMS, particularly by identifying subclonal populations.

To illustrate this validation further, we implemented a DPClust-generated phylogenetic reconstruction. The output of DPClust are CCFs of each mutation that are grouped into clusters, with each cluster having a confidence interval. The tree building tool constructs phylogenies directly from these CCFs by inferring relationships between pairs of clusters, as described here and in the revised methods. We begin by determining whether a pair of clusters is at equal level in the tree (EQ), has an ancestral relationship (GT, greater than or LT, less than) or suggests branching (BRANCH). These labels are established by investigating the overlap of the confidence intervals. EQ is assigned when the intervals overlap, GT when the interval

of cluster a is greater than that of b , LT when b is greater than a , and BRANCH when both GT and LT occur in different samples. The algorithm consists of three main parts. First, identification of a root node. The root node is selected as the cluster that has the highest summed CCF across all samples. Second, a base tree is constructed by assigning clusters with only one possible ancestor already on the tree, which means the cluster must have only one relationship of the type EQ or LT with another cluster. Finally, all remaining clusters are fit onto the base tree, possibly spawning new trees if a cluster fits in multiple positions. This is done by using both the cluster-pair classifications and the cluster CCFs. To test whether a node fits, a variant of the pigeonhole principle is introduced, called the '*weak pigeonhole principle*'. It states that the sum of CCFs of all tree nodes at a particular level cannot exceed 1. The level of a node on the tree is defined by how many steps must be taken to reach the root of the tree, plus 1. For example, the root is level 1 and nodes directly below it are level 2. A parameter leeway is introduced, which is the amount of CCF the sum can go over 1, before the *weak pigeonhole principle* metric is violated. A cluster is placed on the tree when the *weak pigeonhole principle* metric is met. If a cluster fits in multiple places, then new trees are added that represent the different fits. Newly added clusters are placed on all putative trees. Therefore, the number of trees can grow quickly, but typically the solution space is highly constrained, giving rise to a small number of possible trees. This is sample agnostic, clone-based, and demonstrates the branching patterns and clonal composition of LMS patients' tumors.

Having validated that both methods produce the same tree structure, we modified our figures to include details on clonal reconstruction from DPCLust (Main Figures 3-4, Supplementary Figures 21-23, and p. 29, lines 1-23). An example is shown for patient Ab6 - the patient with the most samples sequenced. For each tree, the relative branch length is derived from Treeomics' mutation assignments. The clones (from DPCLust) are displayed as nodes and endpoints.

As demonstrated by both Treeomics and DPCLust, all samples can be traced back to a common ancestral clone (clone 2 in Ab6 below), which harbors the major LMS driver alterations (Homozygous *TP53* mutations and *RB1* LOH). With respect to multiregion samples, the origins of these specimens can be traced back to their most recent common clone (clone 7 in Ab6).

With regards to the shared subclones, to show the various cancer populations that exist across regions and paired samples, we previously used the averaged CCF per sample per cluster. In doing so we highlighted shared major clones and subclones. However, averaging of the CCF did not show shared subclones that do not fully share the same mutation catalogues. To more clearly demonstrate shared subclones between samples, we now only consider entries with CCF values greater than 0 for the calculation. In doing so, we find that within subclones that are present predominantly in one sample (or region), a subset of mutations can be shared in a neighbouring region, suggesting these regions do in fact share subclonal mutations. This is demonstrated in Cluster 16 of patient Ab6 (see below). This cluster is defined by 261 mutations with a CCF of 43% in Ab6MuRe3T and 43 of these mutations are found in Ab6MuRe5T with a CCF of 28%, a neighbouring region on the same focus of this metastatic lesion (see panel A for physical separation). In hindsight, this should

have been our original approach and we thank the reviewer for highlighting this issue. To better highlight these shared variants, we have plotted the averaged CCFs per sample, considering only CCF values greater than 0 (see below in panel C and Supplementary Figures 21-23). To be considered for inclusion in the tree, mutations must represent at least 5% CCF, 5% of the total mutations of the cluster and have at least 5 mutations in that sample. For example, in Cluster 16, Region 5 has 43/291 of total mutations and a CCF of 43%, satisfying all three conditions (pale brown clone in Region 5 below). Conversely, Cluster 16 in Region 4 consists of only 10/291 (3%) of total mutations with a 12% CCF and thus is not included in the tree composition. While only this subclone appears shared between regions, shared subclones are much more prevalent in the Ab11 primary tumor (Dx, Figure 4B). We have annotated the dots in the clonal plot with the number of mutations assigned to the cluster in each sample (Panel C, below). This provides further information on sample composition in the phylogenies. While the manuscript's overall messages are unchanged, we believe these new analyses have refined our understanding of the subclonal relationship between adjacent regions of LMS.

Reviewer #3 (Remarks to the Author):

The authors have addressed all of my comments and concerns.

- 1. One minor comment: It would be helpful to replace "CRL-7518" (which is the ATCC order number) on page 13 with the actual name of the cell line (Hs 789.Sk) and mention that this is a normal skin cell line. Accordingly, the M&M section should be adapted.**

All mentions of CRL-7518 has been replaced with Hs789.sk (page 13, line 16, supplementary figure 15, figure legends). The materials and methods sections have been updated accordingly (page 26, line 31).

REVIEWERS' COMMENTS

Reviewer #2 (Remarks to the Author):

Thank you, the authors have addressed all my concerns.